# Geometric Algebra Transformer

**Johann Brehmer**[*]     **Pim de Haan**[*]     **Sönke Behrends**     **Taco Cohen**

Qualcomm AI Research[†]

{jbrehmer, pim, sbehrend, tacos}@qti.qualcomm.com

## Abstract

Problems involving geometric data arise in physics, chemistry, robotics, computer vision, and many other fields. Such data can take numerous forms, for instance points, direction vectors, translations, or rotations, but to date there is no single architecture that can be applied to such a wide variety of geometric types while respecting their symmetries. In this paper we introduce the Geometric Algebra Transformer (GATr), a general-purpose architecture for geometric data. GATr represents inputs, outputs, and hidden states in the projective geometric (or Clifford) algebra, which offers an efficient 16-dimensional vector-space representation of common geometric objects as well as operators acting on them. GATr is equivariant with respect to $E(3)$, the symmetry group of 3D Euclidean space. As a Transformer, GATr is versatile, efficient, and scalable. We demonstrate GATr in problems from $n$-body modeling to wall-shear-stress estimation on large arterial meshes to robotic motion planning. GATr consistently outperforms both non-geometric and equivariant baselines in terms of error, data efficiency, and scalability.

## 1 Introduction

From molecular dynamics to astrophysics, from material design to robotics, fields across science and engineering deal with geometric data such as positions, shapes, orientations, or directions. The geometric nature of data provides a rich structure: a notion of common operations between geometric types (computing distances between points, applying rotations to orientations, etc.), a well-defined behaviour of data under transformations of a system, and the independence of certain properties of coordinate system choices. When learning from geometric data, incorporating this rich structure into the architecture has the potential to improve the performance.

With this goal, we introduce the *Geometric Algebra Transformer* (GATr), a general-purpose network architecture for geometric data. GATr brings together three key ideas.

**Geometric algebra:** To naturally describe both geometric objects as well as their transformations in three-dimensional space, GATr represents data as multivectors of the projective geometric algebra $\mathbb{G}_{3,0,1}$. Geometric (or Clifford) algebra is an principled yet practical mathematical framework for geometrical computations. The particular algebra $\mathbb{G}_{3,0,1}$ extends the vector space $\mathbb{R}^3$ to 16-dimensional multivectors, which can natively represent various geometric types and $E(3)$ poses. Unlike the $O(3)$ representations popular in geometric deep learning, this algebra can represent data that is not invariant to translations, such as absolute positions.

**Equivariance:** GATr is equivariant with respect to $E(3)$, the symmetry group of three-dimensional space. To this end, we develop several new $E(3)$-equivariant primitives mapping between multivectors, including equivariant linear maps, an attention mechanism, nonlinearities, and normalization layers.

---

[*]Equal contribution.

[†]Qualcomm AI Research is an initiative of Qualcomm Technologies, Inc.

37th Conference on Neural Information Processing Systems (NeurIPS 2023).

**Transformer:** Due to its favorable scaling properties, expressiveness, trainability, and versatility, the Transformer architecture [50] has become the de-facto standard for a wide range of problems. GATr is based on the Transformer architecture, in particular on dot-product attention, and inherits these benefits.

GATr thus combines two lines of research: the representation of geometric objects with geometric algebra [17, 18, 38], popular in computer graphics and physics and recently gaining traction in deep learning [5, 41, 46], and the encoding of symmetries through equivariant deep learning [12]. The result—to the best of our knowledge the first E(3)-equivariant architecture with internal geometric algebra representations[3]—is a versatile network for problems involving geometric data.

We demonstrate GATr in three problems from entirely different fields. In an $n$-body modelling task, we compare GATr to various baselines. We turn towards the task of predicting the wall shear stress in human arteries, demonstrating that GATr scales to realistic problems with meshes of thousands of nodes. Finally, experiment with robotic motion planning, using GATr as the backbone of an E(3)-invariant diffusion model. In all cases, GATr substantially outperforms both non-geometric and equivariant baselines.

Our implementation of GATr is available at `https://github.com/qualcomm-ai-research/geometric-algebra-transformer`.

## 2 Background

**Geometric algebra**  We begin with a brief overview of geometric algebra (GA); for more detail, see e. g. Refs. [17, 18, 38, 41]. Whereas a plain vector space like $\mathbb{R}^3$ allows us to take linear combinations of elements $x$ and $y$ (vectors), a geometric algebra additionally has a bilinear associative operation: the *geometric product*, denoted simply by $xy$. By multiplying vectors, one obtains so-called *multivectors*, which can represent both geometrical *objects* and *operators*. Like vectors, multivectors have a notion of direction as well as magnitude and orientation (sign), and can be linearly combined.

Multivectors can be expanded on a multivector basis, consisting of products of basis vectors. For example, in a 3D GA with orthogonal basis $e_1, e_2, e_3$, a general multivector takes the form

$$x = x_s + x_1 e_1 + x_2 e_2 + x_3 e_3 + x_{12} e_1 e_2 + x_{13} e_1 e_3 + x_{23} e_2 e_3 + x_{123} e_1 e_2 e_3, \tag{1}$$

with real coefficients $(x_s, x_1, \ldots, x_{123}) \in \mathbb{R}^8$. Thus, similar to how a complex number $a + bi$ is a sum of a real scalar and an imaginary number,[4] a general multivector is a sum of different kinds of elements. These are characterized by their dimensionality (grade), such as scalars (grade 0), vectors $e_i$ (grade 1), bivectors $e_i e_j$ (grade 2), all the way up to the *pseudoscalar* $e_1 \cdots e_d$ (grade $d$).

The geometric product is characterized by the fundamental equation $vv = \langle v, v \rangle$, where $\langle \cdot, \cdot \rangle$ is an inner product. In other words, we require that the square of a vector is its squared norm. In an orthogonal basis, where $\langle e_i, e_j \rangle \propto \delta_{ij}$, one can deduce that the geometric product of two different basis vectors is antisymmetric: $e_i e_j = -e_j e_i$[5]. Since reordering only produces a sign flip, we only get one basis multivector per unordered subset of basis vectors, and so the total dimensionality of a GA is $\sum_{i=0}^d \binom{d}{k} = 2^d$. Moreover, using bilinearity and the fundamental equation one can compute the geometric product of arbitrary multivectors.

The symmetric and antisymmetric parts of the geometric product are called the interior and exterior (wedge) product. For vectors $x$ and $y$, these are defined as $\langle x, y \rangle = (xy + yx)/2$ and $x \wedge y \equiv (xy - yx)/2$. The former is indeed equal to the inner product used to define the GA, whereas the latter is new notation. Whereas the inner product computes the similarity, the exterior product constructs a multivector (called a blade) representing the weighted and oriented subspace spanned by the vectors. Both operations can be extended to general multivectors [18].

The final primitive of the GA that we will require is the dualization operator $x \mapsto x^*$. It acts on basis elements by swapping "empty" and "full" dimensions, e. g. sending $e_1 \mapsto e_{23}$.

---

[3]Concurrently to our work, a similar network architecture was studied by Ruhe et al. [40]; we comment on similarities and differences in Sec. 5.

[4]Indeed the imaginary unit $i$ can be thought of as the bivector $e_1 e_2$ in a 2D GA.

[5]The antisymmetry can be derived by using $v^2 = \langle v, v \rangle$ to show that $e_i e_j + e_j e_i = (e_i + e_j)^2 - e_i^2 - e_j^2 = 0$.

| Object / operator | Scalar 1 | Vector $e_0$ | Vector $e_i$ | Bivector $e_{0i}$ | Bivector $e_{ij}$ | Trivector $e_{0ij}$ | Trivector $e_{123}$ | PS $e_{0123}$ |
|---|---|---|---|---|---|---|---|---|
| Scalar $\lambda \in \mathbb{R}$ | $\lambda$ | 0 | 0 | 0 | 0 | 0 | 0 | 0 |
| Plane w/ normal $n \in \mathbb{R}^3$, origin shift $d \in \mathbb{R}$ | 0 | $d$ | $n$ | 0 | 0 | 0 | 0 | 0 |
| Line w/ direction $n \in \mathbb{R}^3$, orthogonal shift $s \in \mathbb{R}^3$ | 0 | 0 | 0 | $s$ | $n$ | 0 | 0 | 0 |
| Point $p \in \mathbb{R}^3$ | 0 | 0 | 0 | 0 | 0 | $p$ | 1 | 0 |
| Pseudoscalar $\mu \in \mathbb{R}$ | 0 | 0 | 0 | 0 | 0 | 0 | 0 | $\mu$ |
| Reflection through plane w/ normal $n \in \mathbb{R}^3$, origin shift $d \in \mathbb{R}$ | 0 | $d$ | $n$ | 0 | 0 | 0 | 0 | 0 |
| Translation $t \in \mathbb{R}^3$ | 1 | 0 | 0 | $\frac{1}{2}t$ | 0 | 0 | 0 | 0 |
| Rotation expressed as quaternion $q \in \mathbb{R}^4$ | $q_0$ | 0 | 0 | 0 | $q_i$ | 0 | 0 | 0 |
| Point reflection through $p \in \mathbb{R}^3$ | 0 | 0 | 0 | 0 | 0 | $p$ | 1 | 0 |

Table 1: Embeddings of common geometric objects and transformations into the projective geometric algebra $\mathbb{G}_{3,0,1}$. The columns show different components of the multivectors with the corresponding basis elements, with $i, j \in \{1, 2, 3\}, j \neq i$, i.e. $ij \in \{12, 13, 23\}$. For simplicity, we fix gauge ambiguities (the weight of the multivectors) and leave out signs (which depend on the ordering of indices in the basis elements).

**Projective geometric algebra**   In order to represent three-dimensional objects as well as arbitrary rotations and translations acting on them, the 3D GA is not enough: as it turns out, its multivectors can only represent linear subspaces passing through the origin as well as rotations around it. A common trick to expand the range of objects and operators is to embed the space of interest (e.g. $\mathbb{R}^3$) into a higher dimensional space whose multivectors represent more general objects and operators in the original space.

In this paper we work with the projective geometric algebra $\mathbb{G}_{3,0,1}$ [17, 38, 41]. Here one adds a fourth *homogeneous coordinate* $x_0 e_0$ to the vector space, yielding a $2^4 = 16$-dimensional geometric algebra. The metric of $\mathbb{G}_{3,0,1}$ is such that $e_0^2 = 0$ and $e_i^2 = 1$ for $i = 1, 2, 3$. As we will explain in the following, in this setup the 16-dimensional multivectors can represent 3D points, lines, and planes, which need not pass through the origin, and arbitrary rotations, reflections, and translations in $\mathbb{R}^3$.

**Representing transformations**   In geometric algebra, a vector $u$ can act as an operator, reflecting other elements in the hyperplane orthogonal to $u$. Since any orthogonal transformation is equal to a sequence of reflections, this allows us to express any such transformation as a geometric product of (unit) vectors, called a (unit) *versor* $u = u_1 \cdots u_k$. Furthermore, since the product of unit versors is a unit versor, and unit vectors are their own inverse ($u^2 = 1$), these form a group called the Pin group associated with the metric. Similarly, products of an even number of reflections form the Spin group. In the projective geometric algebra $\mathbb{G}_{3,0,1}$, these are the double cover[6] of E(3) and SE(3), respectively. We can thus represent any rotation, translation, and mirroring—the symmetries of three-dimensional space—as $\mathbb{G}_{3,0,1}$ multivectors.

In order to apply a versor $u$ to an arbitrary element $x$, one uses the *sandwich product*:

$$\rho_u(x) = \begin{cases} uxu^{-1} & \text{if } u \text{ is even} \\ u\hat{x}u^{-1} & \text{if } u \text{ is odd} \end{cases} \tag{2}$$

Here $\hat{x}$ is the *grade involution*, which flips the sign of odd-grade elements such as vectors and trivectors, while leaving even-grade elements unchanged. Equation 2 thus gives us a linear action (i.e. group representation) of the Pin and Spin groups on the $2^d$-dimensional space of multivectors. The sandwich product is grade-preserving, so this representation splits into a direct sum of representations on each grade.

**Representing 3D objects**   Following Refs. [17, 38, 41], we represent planes with vectors, and require that the intersection of two geometric objects is given by the wedge product of their representations. Lines (the intersection of two planes) are thus represented as bivectors, points (the intersection of three planes) as trivectors. This leads to a duality between objects and operators, where objects are represented like transformations that leave them invariant. Table 1 provides a dictionary of these embeddings. It is easy to check that this representation is consistent with using the sandwich product for transformations.

---

[6]This means that for each element of E(3) there are two elements of Pin(3, 0, 1), e.g. both the vector $v$ and $-v$ represent the same reflection.

**Equivariance**   We construct network layers that are equivariant with respect to E(3), or equivalently its double cover Pin(3, 0, 1). A function $f : \mathbb{G}_{3,0,1} \rightarrow \mathbb{G}_{3,0,1}$ is Pin(3, 0, 1)-equivariant with respect to the representation $\rho$ (or Pin(3, 0, 1)-equivariant for short) if

$$f(\rho_u(x)) = \rho_u(f(x)) \tag{3}$$

for any $u \in \text{Pin}(3, 0, 1)$ and $x \in \mathbb{G}_{3,0,1}$, where $\rho_u(x)$ is the sandwich product defined in Eq. (2).

## 3   The Geometric Algebra Transformer

### 3.1   Design principles and architecture overview

**Geometric inductive bias through geometric algebra representations**   GATr is designed to provide a strong inductive bias for geometric data. It should be able to represent different geometric objects and their transformations, for instance points, lines, planes, translations, rotations, and so on. In addition, it should be able to represent common interactions between these types with few layers, and be able to identify them from little data (while maintaining the low bias of large transformer models). Examples of such common patterns include computing the relative distances between points, applying geometric transformations to objects, or computing the intersections of planes and lines.

Following a body of research in computer graphics, we propose that geometric algebra gives us a language that is well-suited to this task. We use the projective geometric algebra $\mathbb{G}_{3,0,1}$ and use the plane-based representation of geometric structure outlined in the previous section.

**Symmetry awareness through E(3) equivariance**   Our architecture should respect the symmetries of 3D space. Therefore we design GATr to be equivariant with respect to the symmetry group E(3) of translations, rotations, and reflections.

Note that the projective geometric algebra naturally offers a faithful representation of E(3), including translations. We can thus represent objects that transform arbitrarily under E(3), including with respect to translations of the inputs. This is in stark contrast with most E(3)-equivariant architectures, which only use O(3) representations and whose features only transform under rotation and are invariant under translations—and hence can not represent absolute positions like GATr can. Those architectures must handle points in hand-crafted ways, like by canonicalizing w. r. t. the center of mass or by treating the difference between points as a translation-invariant vector.

Many systems will not exhibit the full E(3) symmetry group. The direction of gravity, for instance, often breaks it down to the smaller E(2) group. To maximize the versatility of GATr, we choose to develop a E(3)-equivariant architecture and to include symmetry-breaking as part of the network inputs, similar to how position embeddings break permutation equivariance in transformers.

**Scalability and flexibility through dot-product attention**   Finally, GATr should be expressive, easy to train, efficient, and scalable to large systems. It should also be as flexible as possible, supporting variable geometric inputs and both static scenes and time series.

These desiderata motivate us to implement GATr as a transformer [50], based on attention over multiple objects (similar to tokens in MLP or image patches in computer vision). This choice makes GATr equivariant also with respect to permutations along the object dimension. As in standard transformers, we can break this equivariance when desired (in particular, along time dimensions) through positional embedding.

Like a vanilla transformer, GATr is based on a dot-product attention mechanism, for which heavily optimized implementations exist [14, 32, 37]. We will demonstrate later that this allows us to scale GATr to problems with many thousands of tokens, much further than equivariant architectures based on graph neural networks and message-passing algorithms.

**Architecture overview**   We sketch GATr in Fig. 1. In the top row, we sketch the overall workflow. If necessary, raw inputs are first preprocessed into geometric types. The geometric objects are then embedded into multivectors of the geometric algebra $\mathbb{G}_{3,0,1}$, following the recipe described in Tbl. 1.

The multivector-valued data are processed with a GATr network. We show this architecture in more detail in the bottom row of Fig. 1. GATr consists of $N$ transformer blocks, each consisting

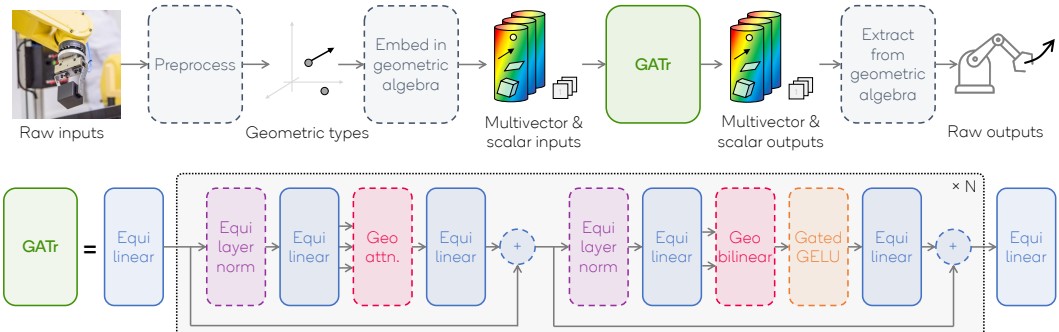

Figure 1: Overview over the GATr architecture. Boxes with solid lines are learnable components, those with dashed lines are fixed.

of an equivariant multivector LayerNorm, an equivariant multivector self-attention mechanism, a residual connection, another equivariant LayerNorm, an equivariant multivector MLP with geometric bilinear interactions, and another residual connection. The architecture is thus similar to a typical transformer [50] with pre-layer normalization [2, 54], but adapted to correctly handle multivector data and be $\mathrm{E}(3)$ equivariant. We describe the individual layers below.

Finally, from the outputs of the GATr network we extract the target variables, again following the mapping given in Tbl. 1.

## 3.2 GATr primitives

**Linear layers**   We begin with linear layers between multivectors. In Appendix A, we show that the equivariance condition of Eq. (3) severely constrains them:

**Proposition 1.** *Any linear map* $\phi : \mathbb{G}_{d,0,1} \to \mathbb{G}_{d,0,1}$ *that is equivariant to* $\mathrm{Pin}(d, 0, 1)$ *is of the form*

$$\phi(x) = \sum_{k=0}^{d+1} w_k \langle x \rangle_k + \sum_{k=0}^{d} v_k e_0 \langle x \rangle_k \tag{4}$$

*for parameters* $w \in \mathbb{R}^{d+2}, v \in \mathbb{R}^{d+1}$. *Here* $\langle x \rangle_k$ *is the blade projection of a multivector, which sets all non-grade-$k$ elements to zero.*

Thus, $\mathrm{E}(3)$-equivariant linear maps between $\mathbb{G}_{3,0,1}$ multivectors can be parameterized with nine coefficients, five of which are the grade projections and four include a multiplication with the homogeneous basis vector $e_0$. We thus parameterize affine layers between multivector-valued arrays with Eq. (4), with learnable coefficients $w_k$ and $v_k$ for each combination of input channel and output channel. In addition, there is a learnable bias term for the scalar components of the outputs (biases for the other components are not equivariant).

**Geometric bilinears**   Equivariant linear maps are not sufficient to build expressive networks. The reason is that these operations allow for only very limited grade mixing, as shown in Prop. 1. For the network to be able to construct new geometric features from existing ones, such as the translation vector between two points, two additional primitives are essential.

The first is the geometric product $x, y \mapsto xy$, the fundamental bilinear operation of geometric algebra. It allows for substantial mixing between grades: for instance, the geometric product of vectors consists of scalars and bivector components. The geometric product is equivariant (Appendix A).

The second geometric primitive we use is derived from the so-called *join*[7] $x, y \mapsto (x^* \wedge y^*)^*$. This map may appear complicated, but it plays a simple role in our architecture: an equivariant map that involves the dual $x \mapsto x^*$. Including the dual in an architecture is essential for expressivity: in $\mathbb{G}_{3,0,1}$, without any dualization it is impossible to represent even simple functions such as the Euclidean

---

[7]Technically, the join has an anti-dual, not the dual, in the output. We leave this detail out for notational simplicity. In our network architecture, it makes no difference for equivariance nor for expressivity whether the anti-dual or dual is used, as any equivariant linear layer can transform between the two.

distance between two points [17]; we show this in Appendix A. While the dual itself is not $\mathrm{Pin}(3,0,1)$-equivariant (w.r.t. $\rho$), the join operation is equivariant to even (non-mirror) transformations. To make the join equivariant to mirrorings as well, we multiply its output with a pseudoscalar derived from the network inputs: $x, y, z \mapsto \mathrm{EquiJoin}(x, y; z) = z_{0123}(x^* \wedge y^*)^*$, where $z_{0123} \in \mathbb{R}$ is the pseudoscalar component of a reference multivector $z$ (see Appendix B).

We define a *geometric bilinear layer* that combines the geometric product and the join of the two inputs as $\mathrm{Geometric}(x, y; z) = \mathrm{Concatenate}_{\mathrm{channels}}(xy, \mathrm{EquiJoin}(x, y; z))$. In GATr, this layer is included in the MLP.

**Nonlinearities and normalization** We use scalar-gated GELU nonlinearities [23] $\mathrm{GatedGELU}(x) = \mathrm{GELU}(x_1)x$, where $x_1$ is the scalar component of the multivector $x$. Moreover, we define an $E(3)$-equivariant LayerNorm operation for multivectors as $\mathrm{LayerNorm}(x) = x/\sqrt{\mathbb{E}_c\langle x, x \rangle}$, where the expectation goes over channels and we use the invariant inner product $\langle \cdot, \cdot \rangle$ of $\mathbb{G}_{3,0,1}$.

**Attention** Given multivector-valued query, key, and value tensors, each consisting of $n_i$ items (or tokens) and $n_c$ channels (key length), we define the $E(3)$-equivariant multivector attention as

$$\mathrm{Attention}(q, k, v)_{i'c'} = \sum_i \mathrm{Softmax}_i \left( \frac{\sum_c \langle q_{i'c}, k_{ic} \rangle}{\sqrt{8 n_c}} \right) v_{ic'}. \tag{5}$$

Here the indices $i, i'$ label items, $c, c'$ label channels, and $\langle \cdot, \cdot \rangle$ is the invariant inner product of the geometric algebra. Just as in the original transformer [50], we thus compute scalar attention weights with a scaled dot product; the difference is that we use the inner product of $\mathbb{G}_{3,0,1}$, which is the regular $\mathbb{R}^8$ dot product on 8 of the 16 dimensions, ignoring the dimensions containing $e_0$.[8] We compute the attention using highly efficient implementations of conventional dot-product attention [14, 32, 37]. As we will demonstrate later, this allows us to scale GATr to systems with many thousands of tokens. We extend this attention mechanism to multi-head self-attention in the usual way.

### 3.3 Extensions

**Auxiliary scalar representations** While multivectors are well-suited to model geometric data, many problems contain non-geometric information as well. Such scalar information may be high-dimensional, for instance in sinusoidal positional encoding schemes. Rather than embedding into the scalar components of the multivectors, we add an auxiliary scalar representation to the hidden states of GATr. Each layer thus has both scalar and multivector inputs and outputs. They have the same batch dimension and item dimension, but may have different number of channels.

This additional scalar information interacts with the multivector data in two ways. In linear layers, we allow the auxiliary scalars to mix with the scalar component of the multivectors. In the attention layer, we compute attention weights both from the multivectors, as given in Eq. (5), and from the auxiliary scalars, using a regular scaled dot-product attention. The two attention maps are summed before computing the softmax, and the normalizing factor is adapted. In all other layers, the scalar information is processed separately from the multivector information, using the unrestricted form of the multivector map. For instance, nonlinearities transform multivectors with equivariant gated GELUs and auxiliary scalars with regular GELU functions. We describe the scalar path of our architecture in more detail in Appendix B.

**Distance-aware dot-product attention** The dot-product attention in Eq. (5) ignores the 8 dimensions involving the basis element $e_0$. These dimensions vary under translations, and thus their straightforward Euclidean inner product violates equivariance. We can, however, extend the attention mechanism to incorporate more components, while still maintaining $E(3)$ equivariance and the computational efficiency of dot-product attention. To this end, we define certain auxiliary, non-linear query features $\phi(q)$ and key features $\psi(k)$ and extend the attention weights in Eq. (5) as $\langle q_{i'c}, k_{ic} \rangle \to \langle q_{i'c}, k_{ic} \rangle + \phi(q_{i'c}) \cdot \psi(k_{ic})$, adapting the normalization appropriately. We define these nonlinear features in Appendix B.

---

[8]We also experimented with attention mechanisms that use the geometric product rather than the dot product, but found a worse performance in practice.

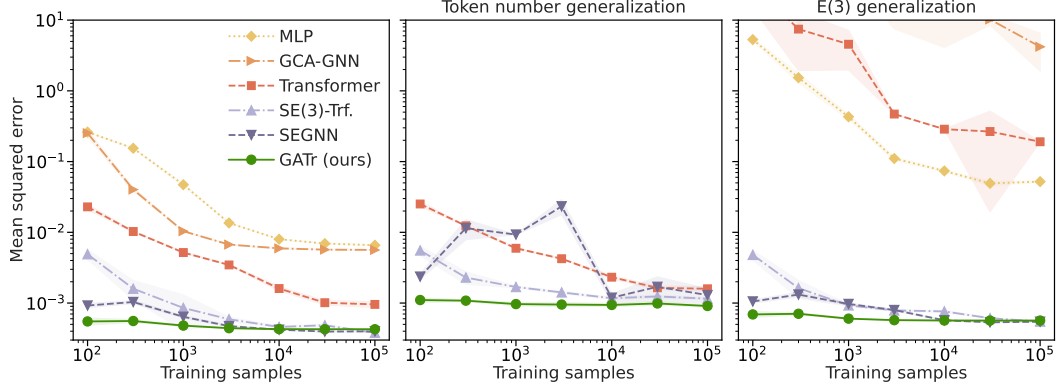

Figure 2: $n$-body dynamics experiments. We show the error in predicting future positions of planets as a function of the training dataset size. Out of five independent training runs, the mean and standard error are shown. **Left**: Evaluating without distribution shift. GATr (——) is more sample efficient than the equivariant SEGNN [6] (——) and SE(3)-Transformer [20] (— ·) and clearly outperforms non-equivariant baselines (·····, —·—, ——). **Middle**: Evaluating on systems with more planets than trained on. Transformer architectures generalize well to different object counts. GCA-GNN has larger errors than the visible range. **Right**: Evaluating on translated data. Because GATr is $E(3)$ equivariant, it generalizes under this domain shift.

Our choice of these nonlinear features not only maintains equivariance, but has a straightforward geometric interpretation. When the trivector components of queries and keys represent 3D points (see Tbl. 1), $\psi(q) \cdot \phi(k)$ is proportional to the pairwise negative squared Euclidean distance. GATr's attention mechanism is therefore directly sensitive to Euclidean distance, while still respecting the highly efficient dot-product attention format.

**Positional embeddings**   GATr assumes the data can be described as a set of items (or tokens). If these items are distinguishable and form a sequence, we encode their position using "rotary positional"[9] embeddings [47] in the auxiliary scalar variables.[10]

**Axial attention**   The architecture is flexible about the structure of the data. In some use cases, there will be a single dimension along which objects are organized, for instance when describing a static scene or the time evolution of a single object. But GATr also supports the organization of a problem along multiple axes, for example with one dimension describing objects and another time steps. In this case, we follow an axial transformer layout [24], alternating between transformer blocks that attend over different dimensions. (The not-attended dimensions in each block are treated like a batch dimension.)

## 4   Experiments

### 4.1   $n$-body dynamics

We start our empirical demonstration of GATr with an $n$-body dynamics problem, on which we compare GATr to a wide range of baselines. Given the masses, initial positions, and velocities of a star and a few planets, the goal is to predict the final position after the system has evolved under Newtonian gravity for 1000 Euler integration time steps. We compare GATr to a Transformer and an MLP, the equivariant SEGNN [6] and SE(3)-Transformer [20], as well as the geometric-algebra-based—but not equivariant—GCA-GNN [41]. The experiment is described in detail in Appendix C.1.

In the left panel of Fig. 2 we show the prediction error as a function of the number of training samples used. GATr clearly outperforms all non-equivariant baselines, including the geometric-algebra-based GCA-GNN. Compared to the equivariant SEGNN and SE(3)-Transformer, GATr is more sample-

---

[9]This terminology, which stems from non-geometric transformers, can be confusing. "Position" here means position in a sequence, not geometric position in 3D. "Rotary" does not refer a rotation of 3D space, but rather to how the position in a sequence is embedded via sinusoids in the scalar channels of keys and queries. Using positional encoding thus does not affect $E(3)$ equivariance.

[10]Since auxiliary scalar representations and multivectors mix in the attention mechanism, the positional embeddings also affect the multivector processing.

efficient, while all three methods reach the same prediction error when trained on enough data. This provides evidence for the usefulness of geometric algebra representations as an inductive bias.

GATr also generalizes robustly out of domain, as we show in the middle and right panels of Fig. 1. When evaluating on a larger number of bodies than trained on, methods that use a softmax over attention weights (GATr, Transformer, $SE(3)$-Transformer) generalize best. Finally, the performance of the $E(3)$-equivariant GATr, SEGNN, and $SE(3)$-Transformer does not drop when evaluated on spatially translated data, while the non-equivariant baselines fail in this setting.

## 4.2 Wall-shear-stress estimation on large meshes of human arteries

Next, we turn towards a realistic experiment involving more complex geometric objects. We study the prediction of the wall shear stress exerted by the blood flow on the arterial wall, an important predictor of aneurysms. While the wall shear stress can be computed with computational fluid dynamics, simulating a single artery can take many hours, and efficient neural surrogates can have substantial impact. However, training such neural surrogates is challenging, as meshes are large (around 7000 nodes in our data) and datasets typically small (we work with 1600 training meshes).

We train GATr on a dataset of arterial meshes and simulated wall shear stress published by Suk et al. [48]. They are compared to a Transformer and to the results reported by Suk et al. [48], including the equivariant GEM-CNN [15] and the non-equivariant PointNet++ [36].[11] See Appendix C.2 for experiment details.

The results are shown in Fig. 3. On non-canonicalized data, with randomly rotated meshes, GATr improves upon all previous methods and sets a new state of the art.

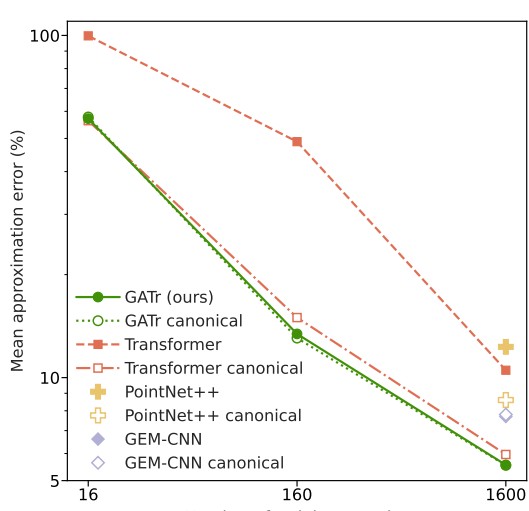

Figure 3: Arterial wall-shear-stress estimation [48]. We show the mean approximation error (lower is better) as a function of training dataset size, reporting results both on randomly oriented training and test samples (solid markers) and on a version of the dataset in which all artery meshes are canonically oriented (hollow markers). Without canonicalization, GATr (——) predicts wall shear stress more precisely and is more sample-efficient than the baselines.

We also experiment with canonicalization: rotating the arteries such that blood always flows in the same direction. This helps the Transformer to be almost competitive with GATr. However, canonicalization is only feasible for relatively straight arteries as in this dataset, not in more complex scenarios with branching and turning arteries. We find it likely that GATr will be more robust in such scenarios.

## 4.3 Robotic planning through invariant diffusion

In our third experiment, we show how GATr defines an $E(3)$-invariant diffusion model, that it can be used for model-based reinforcement learning and planning, and that this combination is well-suited to solve robotics problems. We follow Janner et al. [27], who propose to treat learning a world model and planning within that model as a unified generative modeling problem. After training a diffusion model [45] on offline trajectories, one can use it in a planning loop, sampling from it conditional on the current state, desired future states, or to maximize a given reward, as needed.

We use a GATr model as the denoising network in a diffusion model and to use it for planning. We call this combination *GATr-Diffuser*. Combining the equivariant GATr with an invariant base density defines an $E(3) \times S_n$-invariant diffusion model [29].

GATr-Diffuser is demonstrated on the problem of a Kuka robotic gripper stacking blocks using the "unconditional" environment introduced by Janner et al. [27]. We train GATr-Diffuser on the offline

---

[11]We also tried to run SEGNN [5] and $SE(3)$-Transformer [20] as additional baselines, but were not able to fit them into memory on this task.

trajectory dataset published with that paper and then use it for planning, following the setup of Janner et al. [27]. We compare our GATr-Diffuser model to a reproduction of the original Diffuser model and a new Transformer backbone for the Diffuser model. In addition, we show the published results of Diffuser [27], the equivariant EDGI [7], and the offline RL algorithms CQL [30] and BCQ [21] as published in Ref. [27]. The problem and hyperparameters are described in detail in Appendix C.3.

As shown in Fig. 4, GATr-Diffuser solves the block-stacking task better than all baselines. It is also clearly more sample-efficient, matching the performance of a Diffuser model or Transformer trained on the full dataset even when training only on 1% of the trajectories.

### 4.4   Scaling

Finally, we study GATr's computational requirements and scaling. We measure the memory usage and compute time of forward and backward passes on synthetic data as a function of the number of items. GATr is compared to a

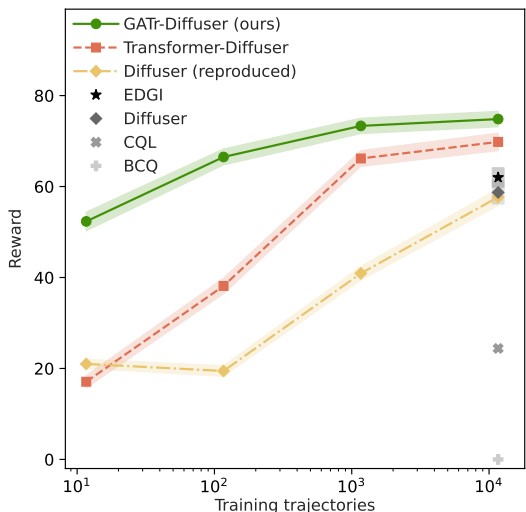

Figure 4: Diffusion-based robotic planning. We show normalized rewards (higher is better) as a function of training dataset size. GATr (━━) is more successful at block stacking and more sample-efficient than the baselines, including the original Diffuser [27] (━·) and our modification of it based on a Transformer (━ ━). In grey, we show results reported by Brehmer et al. [7] and Janner et al. [27].

Transformer, to SEGNN [6] in the official implementation, and to the SE(3)-Transformer [20]; for the latter we use the highly optimized implementation by Milesi [35]. We choose hyperparameters for the four architectures such that they have the same depth and width and require that the methods allow all items to interact at each step (i. e. fully connected graphs). Because of the fundamental differences between the architectures, it is impossible to find fully equivalent settings; our results should thus be interpreted with care. The details of our experiment are described in Appendix C.4.

We show the results in Fig. 5. For few tokens, GATr is slower than a Transformer. However, this difference is partially due to the low utilization of the GPUs in this test; GATr is closer to the Transformer when using larger batch sizes or when pre-compiling the computation graph.

For larger problems, compute and memory are dominated by the pairwise interactions in the attention mechanism. Here GATr and the baseline Transformer perform on par, as they use the same efficient implementation of dot-product attention. In terms of memory, both scale linearly in the number of tokens, while the equivariant baselines scale quadratically. In terms of time, all methods scale quadratically, but the equivariant baselines have a worse prefactor than GATr and the Transformer. All in all, we can easily scale GATr to tens of thousands of tokens, while the equivariant baselines run out of memory two orders of magnitude earlier.

## 5   Related work

**Geometric algebra**    Geometric (or Clifford) algebra was first conceived in the 19th century [10, 22] and has been used widely in quantum physics [16, 34]. Recently, it has found new popularity in computer graphics [18]. In particular,

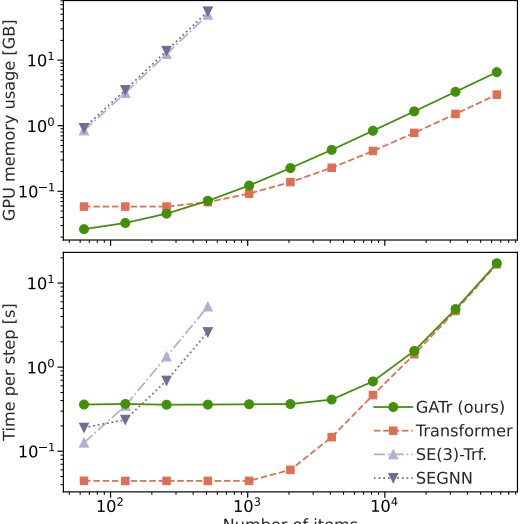

Figure 5: Computational cost and scaling. We measure peak GPU memory usage (top) and wall time (bottom) per combined forward and backward pass as a function of the number of items in synthetic data. Despite some compute overhead, GATr (━━) scales just like the Transformer (━ ━) and orders of magnitude more favorably than the equivariant baselines (━·, ━ ━).

the projective geometric algebra used in this work [17, 38] and a conformal model [18] are suitable for 3D computations.

Geometric algebras have been used in machine learning in various ways. Spellings [46] use $\mathbb{G}_{3,0,0}$ geometric products to compute rotation-invariant features from small point clouds. Unlike us, they do not learn internal geometric representations.

Our work was inspired by Brandstetter et al. [5] and Ruhe et al. [41]. These works also use multivector features (the latter even of the same $\mathbb{G}_{3,0,1}$), and process them with operations such as the geometric / sandwich product, Clifford Fourier transforms and Clifford convolutions. The main difference to our work is that GATr is $E(3)$ equivariant, while both of these works are not. We compare to the GCA-GNN network from Ref. [41] in our experiments.

Concurrently to this publication, Ruhe et al. [40] also study equivariant, geometric algebra–based architectures. While some of our and their findings overlap, there are several differences: They develop theory for generic geometric algebras of the form $\mathbb{G}_{p,q,r}$, while we focus on the projective algebra $\mathbb{G}_{3,0,1}$ with its faithful $E(3)$ representations, with our theory also applicable to the group $E(n)$ and the algebras $\mathbb{G}_{n,0,0}$ and $\mathbb{G}_{n,0,1}$. Ruhe et al. [40] also finds the linear maps of Eq. (4), but does not discuss the join or distance-aware dot-product, which we found to be essential for performance in the projective algebra. Moreover, they propose an MLP-like architecture and use it in a message-passing graph network, while our GATr is a Transformer.

**Equivariant deep learning** Equivariance to symmetries [11] is the primary design principle in modern geometric deep learning [8]. Equivariant networks have been applied successfully in areas such as medical imaging [31, 53] and robotics [7, 26, 44, 51, 52, 55], and are ubiquitous in applications of machine learning to physics and chemistry [1, 3, 4, 19, 28, 33].

In recent years, a number of works have investigated equivariant Transformer and message-passing architectures [3, 4, 6, 19, 20, 39, 42, 49]. These works are generally more limited in terms of the types of geometric quantities they can process compared to our multivector features. Furthermore, our architecture is equivariant to the full group of Euclidean transformations, whereas previous works focus on $SO(3)$ equivariance.

## 6 Discussion

We introduced the Geometric Algebra Transformer (GATr), a general-purpose architecture for geometric data, and implemented it at `https://github.com/qualcomm-ai-research/geometric-algebra-transformer`. We argued and demonstrated that GATr effectively combines structure and scalability.

GATr incorporates geometric *structure* by representing data in projective geometric algebra, as well as through $E(3)$ equivariance. Unlike most equivariant architectures, GATr features faithful $E(3)$ representations, including absolute positions and equivariance with respect to translations. Empirically, GATr outperforms non-geometric, equivariant, and geometric algebra–based non-equivariant baselines across three experiments.

At the same time, GATr *scales* much better than most geometric networks. This is because GATr is a Transformer and computes pairwise interactions through dot-product attention. Using recent efficient attention implementations, we demonstrated that we can scale GATr to systems with many thousands of tokens and fully connected interactions. For few tokens and small batch sizes, GATr has some computational overhead, which we hope to address in future implementations.

One drawback of our approach is that since geometric algebra is not widely known yet, it may present an obstacle to understanding the details of the method. However, given a dictionary for embeddings of common objects and a library of primitives that act on them, *using* this framework is no more difficult than using typical neural network layers grounded in linear algebra. Another potential downside is that GATr is not yet shown to be a universal approximator, which is an interesting direction for future work.

Given the promising results presented in this work, we look forward to further study the potential of Geometric Algebra Transformers in problems from molecular dynamics to robotics.

**Acknowledgements**   We would like to thank Joey Bose, Johannes Brandstetter, Gabriele Cesa, Steven De Keninck, Daniel Dijkman, Leo Dorst, Mario Geiger, Jonas Köhler, Ekdeep Singh Lubana, Evgeny Mironov, and David Ruhe for generous guidance regarding geometry, enthusiastic encouragement on equivariance, and careful counsel concerning computing complications.

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

# A Theoretical results

In this section, we state or prove several properties of equivariant maps between geometric algebras that we use in the construction of GATr.

The grade involution is a linear involutive bijection $\widehat{\cdot} : \mathbb{G}_{n,0,r} : \mathbb{G}_{n,0,r}$, which sends a $k$-blade $x$ to $\widehat{x} = (-1)^k x$. Note that this is an algebra automorphism $\widehat{xy} = \widehat{x}\widehat{y}$, and also an $\wedge$-algebra automorphism. The reversal in a linear involutive bijection $\widetilde{\cdot} : \mathbb{G}_{n,0,r} : \mathbb{G}_{n,0,r}$ which sends a $k$-blade $x = x_1 \wedge x_2 \wedge ... \wedge x_k$ to the reverse: $\widetilde{x} = x_k \wedge ... \wedge x_2 \wedge x_1 = \pm x$ with $+x$ if $k \in \{0, 1, 4, 5, ..., 8, 9, ...\}$ and $-x$ otherwise. Note that this is an anti-automorphism (contravariant functor): $\widetilde{xy} = \widetilde{y}\widetilde{x}$.

Here we denote the sandwich action of $u \in \text{Pin}(n, 0, r)$ on a multivector $x$ not as $\rho_u(x)$, but as $u[x]$. For odd $u$, $u[x] = u\widehat{x}u^{-1}$, while for even $u$, $u[x] = uxu^{-1}$. The sandwich action is linear by linearity of the $\widehat{\cdot}$ and bilinearity of the geometric product. Furthermore, note that for any particular $u \in \text{Pin}(n, 0, r)$, the action is a geometric algebra homomorphism: $u[ab] = u\widehat{ab}u^{-1} = u\widehat{a}u^{-1}u\widehat{b}u^{-1} = u[a]u[b]$. By linearity and a symmetrization argument [18, Sec 7.1], one can show that it also a $\wedge$-algebra homomorphism (outermorphism): $u[a \wedge b] = u[a] \wedge u[b]$.

Let $l \geq k$. Given a $k$-vector $a$ and $l$-vector $b$, define the *left contraction* as $a \rfloor b := \langle ab \rangle_{l-k}$, which is a $l - k$-vector. For $k = 1$, and $b$ a blade $b = b_1 \wedge ... \wedge b_l$. Geometrically, $a \rfloor b$ is the projection of $a$ to the space spanned by the vectors $b_i$. Thus we have that $a \rfloor b = 0 \iff \forall i, \langle a, b_i \rangle = 0$ [18, Sec 3.2.3], in which case we define $a$ and $b$ to be *orthogonal*. In particular, two vectors $a, b$ are orthogonal if their inner product is zero. Futhermore, we define a vector $a$ to be *tangential* to blade $b$ if $a \wedge b = 0$.

In the projective algebra, a blade $x$ is defined to be *ideal* if it can be written as $x = e_0 \wedge y$ for another blade $y$.

## A.1 Linear maps

We begin with Pin-equivariant linear maps. After some technical lemmata, we prove the most general form of linear equivariant maps in the Euclidean geometric algebra $\mathbb{G}_{n,0,0}$, and then also in projective geometric algebra $\mathbb{G}_{n,0,1}$.

**Proposition 2.** *The grade projection $\langle \cdot \rangle_k$ is equivariant [18, Sec 13.2.3].*

*Proof.* Choose an $l$-blade $x = a_1 \wedge a_2 \wedge ... \wedge a_l$. Let $u$ be a 1-versor. As the action $u$ is an outermorphism, $u[x] = u[a_1] \wedge ... \wedge u[a_l]$ is an $l$-blade. Now if $l \neq k$, then $\langle x \rangle_k = 0$ and thus $u[\langle x \rangle_k] = \langle u[x] \rangle_k$. If $l = k$, then $\langle x \rangle_k = x$ and thus $u[\langle x \rangle_k] = \langle u[x] \rangle_k$. As the grade projection is linear, equivariance extends to any multivector. $\square$

**Proposition 3.** *The following map is equivariant: $\phi : \mathbb{G}_{3,0,1} \to \mathbb{G}_{3,0,1} : x \mapsto e_0 x$.*

*Proof.* Let $u$ be a 1-versor, then $u$ acts on a multivector as $x \mapsto u[x] = u\hat{x}u^{-1}$, where $\hat{x}$ is the grade involution. Note that $e_0$ is invariant: $u[e_0] = -ue_0u^{-1} = e_0uu^{-1} = e_0$, where $ue_0 = -e_0u$ because $u$ and $e_0$ are orthogonal: $ue_0 = \langle u, e_0 \rangle + u \wedge e_0 = -e_0 \wedge u = -e_0 u$. Then $\phi$ is equivariant, as the action is an algebra homomorphism: $u[\phi(x)] = u[e_0x] = u\widehat{e_0x}u^{-1} = u\hat{e}_0u^{-1}u\hat{x}u^{-1} = u[e_0]u[x] = e_0u[x] = \phi(u[x])$. It follows that $\phi$ is also equivariant to any product of vectors, i.e. any versor $u$. $\square$

**Euclidean geometric algebra** Before constructing the most general equivariant linear map between multivectors in projective geometric algebra, we begin with the Euclidean case $\mathbb{G}_{n,0,0}$.

**Theorem 1** (Cartan-Dieudonné). *Every orthogonal transformation of an $n$-dimensional space can be decomposed into at most $n$ reflections in hyperplanes.*

*Proof.* This theorem is proven in Roelfs and De Keninck [38]. $\square$

**Lemma 1.** *In the $n$-dimensional Euclidean geometric algebra $\mathbb{G}_{n,0,0}$, the group $\text{Pin}(n, 0, 0)$ acts transitively on the space of $k$-blades of norm $\lambda \in \mathbb{R}^{>0}$.*

*Proof.* As the Pin group preserves norm, choose $\lambda = 1$ without loss of generality. Any $k$-blade $x$ of unit norm can be written by Gram-Schmidt factorization as the wedge product of $k$ orthogonal vectors of unit norm $x = v_1 \wedge v_2 \wedge ... \wedge v_k$. Consider another $k$-blade $y = w_1 \wedge w_2 \wedge ... \wedge w_k$ with $w_i$ orthonormal. We'll construct a $u \in \text{Pin}(n, 0, 0)$ such that $u[x] = y$.

Choose $n - k$ additional orthonormal vectors $v_{k+1}, ..., v_n$ and $w_{k+1}, .., .w_n$ to form orthonormal bases. Then, there exists a unique orthogonal transformation $\mathbb{R}^n \to \mathbb{R}^n$ that maps $v_i$ into $w_i$ for all $i \in \{1, ..., n\}$. By the Cartan-Dieuodonné theorem 1, this orthogonal transformation can be expressed as the product of reflections, thus there exists a $u \in \text{Pin}(n, 0, 0)$ such that $u[v_i] = w_i$. As the $u$ action is a $\wedge$-algebra homomorphism ($u[a \wedge b] = u[a] \wedge u[b]$, for any multivectors $a, b$), we have that $u[x] = y$. $\qquad\square$

**Lemma 2.** *In the Euclidean ($r = 0$) or projective ($r = 1$) geometric algebra $\mathbb{G}_{n,0,r}$, let $x$ be a $k$-blade. Let $u$ be a 1-versor. Then $u[x] = x \iff u \rfloor x = 0$ and $u[x] = -x \iff u \wedge x = 0$.*

*Proof.* Let $x$ be a $k$-blade and $u$ a vector of unit norm. We can decompose $u$ into $u = t + v$ with $t \wedge x = 0$ (the part tangential to the subspace of $x$) and $v \rfloor x = 0$ (the normal part). This decomposition is unique unless $x$ is ideal in the projective GA, in which case the $e_0$ component of $u$ is both normal and tangential, and we choose $t$ Euclidean.

In either case, note the following equalities: $xt = (-1)^{k-1}tx$; $xv = (-1)^k vx$; $vt = -tv$ and note $\nexists \lambda \neq 0$ such that $vtx = \lambda x$, which can be shown e.g. by picking a basis. Then:

$$u[x] = (-1)^k(t+v)x(t+v) = (t+v)(-t+v)x = (-\|t\|^2 + \|v\|^2)x - 2vtx$$

We have $u[x] \propto x \iff vtx = 0$. If $x$ is not ideal, this implies that either $v = 0$ (thus $u \wedge x = 0$ and $u[x] = -x$) or $t = 0$ (thus $u \rfloor x = 0$ and $u[x] = x$). If $x$ is ideal, this implies that either $v \propto e_0$ (thus $u \wedge x = 0$ and $u[x] = -x$) or $t = 0$ (thus $u \rfloor x = 0$ and $u[x] = x$). $\qquad\square$

**Lemma 3.** *Let $r \in \{0, 1\}$. Any linear $\text{Pin}(n, 0, r)$-equivariant map $\phi : \mathbb{G}_{n,0,r} \to \mathbb{G}_{n,0,r}$ can be decomposed into a sum of equivariant maps $\phi = \sum_{lkm} \phi_{lkm}$, with $\phi_{lkm}$ equivariantly mapping $k$-blades to $l$-blades. If $r = 0$ (Euclidean algebra) or $k < n + 1$, such a map $\phi_{lkm}$ is defined by the image of any one non-ideal $k$-blade, like $e_{12...k}$. Instead, if $r = 1$ (projective algebra) and $k = n + 1$, then such a map is defined by the image of a pseudoscalar, like $e_{01...n}$.*

*Proof.* The $\text{Pin}(n, 0, r)$ group action maps $k$-vectors to $k$-vectors. Therefore, $\phi$ can be decomposed into equivariant maps from grade $k$ to grade $l$: $\phi(x) = \sum_{lk} \phi_{lk}(\langle x \rangle_k)$, with $\phi_{lk}$ having $l$-vectors as image, and all $k'$-vectors in the kernel, for $k' \neq k$. Let $x$ be an non-ideal $k$-blade (or pseudoscalar if $k = n + 1$). By lemmas 1 and 4, in both Euclidean and projective GA, the span of the $k$-vectors in the orbit of $x$ contains any $k$-vector. So $\phi_{lk}$ is defined by the $l$-vector $y = \phi_{lk}(x)$. Any $l$-vector can be decomposed as a finite sum of $l$-blades: $y = y_1 + ...y_M$. We can define $\phi_{lkm}(x) = y_m$, extended to all $l$-vectors by equivariance, and note that $\phi_{lk} = \sum_m \phi_{lkm}$. $\qquad\square$

**Proposition 4.** *For an $n$-dimensional Euclidean geometric algebra $\mathbb{G}_{n,0,0}$, any linear endomorphism $\phi : \mathbb{G}_{n,0,0} \to \mathbb{G}_{n,0,0}$ that is equivariant to the $\text{Pin}(n, 0, 0)$ group (equivalently to $O(n)$) is of the type $\phi(x) = \sum_{k=0}^{n} w_k \langle x \rangle_k$, for parameters $w \in \mathbb{R}^{n+1}$.*

*Proof.* By decomposition of Lemma 3, let $\phi$ map from $k$-blades to $l$-blades. Let $x$ be a $k$-blade. Let $u$ be a 1-versor. By Lemma 2, if $u$ is orthogonal to $x$, $u[\phi(x)] = \phi(u[x]) = \phi(x)$ and $u$ is also orthogonal to $\phi(x)$. If $u \wedge x = 0$, then $u[\phi(x)] = \phi(u[x]) = \phi(-x) = -\phi(x)$ and $u \wedge \phi(x) = 0$. Thus any vector in $x$ is in $\phi(x)$ and any vector orthogonal to $x$ is orthogonal to $\phi(x)$, this implies $\phi(x) = w_k x$, for some $w_k \in \mathbb{R}$. By Lemma 3, we can extend $\phi$ to $\phi(y) = w_k y$ for any $k$-vector $y$. $\qquad\square$

**Projective geometric algebra** How about equivariant linear maps in *projective* geometric algebra? The degenerate metric makes the derivation more involved, but in the end we will arrive at a result that is only slightly more general.

**Lemma 4.** *The Pin group of the projective geometric algebra, $\text{Pin}(n, 0, 1)$, acts transitively on the space of $k$-blades with positive norm $\|x\| = \lambda > 0$. Additionally, the group acts transitively on the space of zero-norm $k$-blades of the form $x = e_0 \wedge y$ (called ideal blades), with $\|y\| = \kappa$.*

*Proof.* Let $x = x_1 \wedge ... \wedge x_k$ be a $k$-blade with positive norm $\lambda$. All vectors $x_i$ can be written as $x_i = v_i + \delta_i e_0$, for a nonzero Euclidean vector $v_i$ (meaning with no $e_0$ component) and $\delta_i \in \mathbb{R}$, because if $v_i = 0$, the norm of $x$ would have been 0. Orthogonalize them as $x_2' = x_2 - \langle x_1, x_2 \rangle x_1$, etc., resulting in $x = x_1' \wedge \cdots \wedge x_k'$ with $x_i' = v_i' + \delta_i' e_0$ with orthogonal $v_i'$.

Define the translation $t = 1 + \frac{1}{2} \sum_i \delta_i' e_0 \wedge v_i'$, which makes $x'$ Euclidean: $t[x'] = v_1' \wedge ... \wedge v_k'$. By Lemma 1, the Euclidean Pin group $\mathrm{Pin}(n, 0, 0)$, which is a subgroup of $\mathrm{Pin}(n, 0, 1)$, acts transitively on Euclidean $k$-blades of a given norm. Thus, in the projective geometric algebra $\mathrm{Pin}(n, 0, 1)$, any two $k$-blades of equal positive norm $\lambda$ are related by a translation to the origin and then a $\mathrm{Pin}(n, 0, 0)$ transformation.

For the ideal blades, let $x = e_0 \wedge y$, with $\|y\| = \kappa$. We take $y$ to be Euclidean without loss of generality. For any $g \in \mathrm{Pin}(n, 0, 1)$, $g[e_0] = e_0$, so $g[x] = e_0 \wedge g[y]$. Consider another $x' = e_0 \wedge y'$ with $\|y'\| = \kappa$ and taking $y'$ Euclidean. As $\mathrm{Pin}(n, 0, 0)$ acts transitively on Euclidean $(k-1)$-blades with norm $\kappa$, let $g \in \mathrm{Pin}(n, 0, 0)$ such that $g[y] = y'$. Then $g[x] = x'$. $\qquad\square$

We can now construct the most general equivariant linear map between projective geometric algebras, a key ingredient for GATr:

**Proposition 5.** *For the projective geometric algebra $\mathbb{G}_{n,0,1}$, any linear endomorphism $\phi : \mathbb{G}_{n,0,1} \to \mathbb{G}_{n,0,1}$ that is equivariant to the group $\mathrm{Pin}(n, 0, r)$ (equivalently to $E(n)$) is of the type $\phi(x) = \sum_{k=0}^{n+1} w_k \langle x \rangle_k + \sum_{k=0}^{n} v_k e_0 \langle x \rangle_k$, for parameters $w \in \mathbb{R}^{n+2}, v \in \mathbb{R}^{n+1}$.*

*Proof.* Following Lemma 3, decompose $\phi$ into a linear equivariant map from $k$-blades to $l$-blades. For $k < n + 1$, let $x = e_{12...k}$. Then following Lemma 2, for any $1 \le i \le k$, $e_i \wedge x = 0$, $e_i[x] = -x$, and $e_i[\phi(x)] = \phi(e_i[x]) = \phi(-x) = -\phi(x)$ and thus $e_i \wedge \phi(x) = 0$. Therefore, we can write $\phi(x) = x \wedge y_1 \wedge ... \wedge y_{l-k}$, for $l - k$ vectors $y_j$ orthogonal to $x$.

Also, again using Lemma 2, for $k < i \le n$, $e_i \rfloor x = 0 \implies e_i[\phi(x)] = \phi(x) \implies e_i \rfloor \phi(x) = 0 \implies \forall i, \langle e_i, y_j \rangle = 0$. Thus, $y_j$ is orthogonal to all $e_i$ with $1 \le i \le n$. Hence, $l = k$ or $l = k + 1$ and $y_1 \propto e_0$.

For $k = n + 1$, let $x = e_{012...k}$. By a similar argument, all invertible vectors $u$ tangent to $x$ must be tangent to $\phi(x)$, thus we find that $\phi(x) = x \wedge y$ for some blade $y$. For any non-zero $\phi(x)$, $y \propto 1$, and thus $\phi(x) \propto x$. By Lemma 3, by equivariance and linearity, this fully defines $\phi$. $\qquad\square$

## A.2 Bilinear maps

Next, we turn towards bilinear operations. In particular, we show that the geometric product and the join are equivariant.

For the geometric product, equivariance is straightforward: Any transformation $u \in \mathrm{Pin}(n, 0, r)$, gives a homomorphism of the geometric algebra, as for any multivectors $x, y$, $u[xy] = u \widehat{xy} u^{-1} = u \widehat{x} \widehat{y} u^{-1} = u \widehat{x} u^{-1} u \widehat{y} u^{-1} = u[x] u[y]$. The geometric product is thus equivariant.

**Dual and join in Euclidean algebra** For the join and the closely related dual, we again begin with the Euclidean geometric algebra, before turning to the projective case later.

The role of the dual is to have a bijection $\cdot^* : \mathbb{G}_{n,0,0} \to \mathbb{G}_{n,0,0}$ that maps $k$-vectors to $(n-k)$-vectors. For the Euclidean algebra, with a choice of pseudoscalar $\mathcal{I}$, we can define a dual as:

$$x^* = x \mathcal{I}^{-1} = x \tilde{\mathcal{I}} \tag{6}$$

This dual is bijective, and involutive up to a sign: $(y^*)^* = y \tilde{\mathcal{I}} \tilde{\mathcal{I}} = \pm y$, with $+y = 1$ for $n \in \{1, 4, 5, 8, 9, ...\}$ and $-y$ for $n \in \{2, 3, 6, 7, ...\}$. We choose $\tilde{\mathcal{I}}$ instead of $\mathcal{I}$ in the definition of the dual so that given $n$ vectors $x_1, ..., x_n$, the dual of the multivector $x = x_1 \wedge ... x_n$, is given by the scalar of the oriented volume spanned by the vector. We denote the inverse of the dual as $x^{-*} = x \mathcal{I}$. Expressed in a basis, the dual yields the complementary indices and a sign. For example, for $n = 3$ and $\mathcal{I} = e_{123}$, we have $(e_1)^* = -e_{23}$, $(e_{12})^* = e_3$.

Via the dual, we can define the bilinear join operation, for multivectors $x, y$:

$$x \vee y := (x^* \wedge y^*)^{-\star} = ((x \tilde{\mathcal{I}}) \wedge (y \tilde{\mathcal{I}})) \mathcal{I} .$$

**Lemma 5.** *In Euclidean algebra* $\mathbb{G}_{n,0,0}$, *the join is* $\mathrm{Spin}(n,0,0)$ *equivariant. Furthermore, it is* $\mathrm{Pin}(n,0,0)$ *equivariant if and only if $n$ is even.*

*Proof.* The join is equivariant to the transformations from the group $\mathrm{Spin}(n,0,0)$, which consists of the product of an even amount of unit vectors, because such transformations leave the pseudoscalar $\mathcal{I}$ invariant, and the operation consists otherwise of equivariant geometric and wedge products.

However, let $e_{12\ldots n} = \mathcal{I} \in \mathrm{Pin}(n,0,0)$ be the point reflection, which negates vectors of odd grades by the grade involution: $\mathcal{I}[x] = \hat{x}$. Let $x$ be a $k$-vector and $y$ an $l$-vector. Then $x \vee y$ is a vector of grade $n - ((n-k) + (n-l)) = k + l - n$ (and zero if $k + l < n$). Given that the join is bilinear, the inputs transform as $(-1)^{k+l}$ under the point reflection, while the transformed output gets a sign $(-1)^{k+l-n}$. Thus for odd $n$, the join is not $\mathrm{Pin}(n,0,0)$ equivariant. $\quad\square$

To address this, given a pseudoscalar $z = \lambda\mathcal{I}$, we can create an equivariant Euclidean join via:

$$\mathrm{EquiJoin}(x, y, z = \lambda\mathcal{I}) := \lambda(x \vee y) = \lambda(x^* \wedge y^*)^{-*}. \tag{7}$$

**Proposition 6.** *In Euclidean algebra* $\mathbb{G}_{n,0,0}$, *the equivariant join* $\mathrm{EquiJoin}$ *is* $\mathrm{Pin}(n,0,0)$ *equivariant.*

*Proof.* The $\mathrm{EquiJoin}$ is a multilinear operation, so for $k$-vector $x$ and $l$-vector $y$, under a point reflection the input gets a sign $(-1)^{k+l+n}$ while the output is still a $k + l - n$ vector and gets sign $(-1)^{k+l-n}$. These signs differ by even $(-1)^{2n} = 1$ and thus $\mathrm{EquiJoin}$ is $\mathrm{Pin}(n,0,1)$-equivariant. $\quad\square$

We prove two equalities of the Euclidean join which we use later.

**Lemma 6.** *In the algebra* $\mathbb{G}_{n,0,0}$, *let $v$ be a vector and $x, y$ be multivectors. Then*

$$v \rfloor (x \vee y) = (v \rfloor x) \vee y \tag{8}$$

*and*

$$x \vee (v \rfloor y) = -(-1)^n \widehat{v \rfloor x} \vee y. \tag{9}$$

*Proof.* For the first statement, let $a$ be a $k$-vector and $b$ an $l$-vector. Then note the following two identities:

$$a \vee b = \langle a^* b \tilde{\mathcal{I}} \rangle_{2n-k-l} \mathcal{I} = \langle a^* b \rangle_{n-(2n-k-l)} \tilde{\mathcal{I}} \mathcal{I} = \langle a^* b \rangle_{k+l-n} = a^* \rfloor b,$$
$$(v \rfloor a)^* = \langle va \rangle_{k-1} \tilde{\mathcal{I}} = \langle va \tilde{\mathcal{I}} \rangle_{n-k+1} = \langle va^* \rangle_{n-k+1} = v \rfloor (a^*).$$

Combining these and the associativity of $\rfloor$ gives:

$$(v \rfloor a) \vee b = (v \rfloor a)^* \rfloor b = v \rfloor (a^*) \rfloor b = v \rfloor (a \vee b)$$

For the second statement, swapping $k$-vector $a$ and $l$-vector $b$ incurs $a \vee b = (a^* \wedge b^*)^{-*} = (-1)^{(n-k)(n-l)}(b^* \wedge a^*)^{-*} = (-1)^{(n-k)(n-l)}(b \vee a)$. Then we get:

$$\begin{aligned}
a \vee (v \rfloor b) &= (-1)^{(n-k)(n-l-1)}(v \rfloor b) \vee a \\
&= (-1)^{(n-k)(n-l-1)} v \rfloor (b \vee a) \\
&= (-1)^{(n-k)(n-l-1)+(n-k)(n-l)} v \rfloor (a \vee b) \\
&= (-1)^{(n-k)(n-l-1)+(n-k)(n-l)}(v \rfloor a) \vee b \\
&= (-1)^{(n-k)(2n-2l-1)}(v \rfloor a) \vee b \\
&= (-1)^{k-n}(v \rfloor a) \vee b \\
&= -(-1)^{k-1-n}(v \rfloor a) \vee b \\
&= -(-1)^n \widehat{(v \rfloor a)} \vee b.
\end{aligned}$$

This generalizes to multivectors $x, y$ by linearity. $\quad\square$

**Dual and join in projective algebra** For the projective algebra $\mathbb{G}_{n,0,1}$ with its degenerate inner product, the dual definition of Eq. 6 unfortunately does not yield a bijective dual. For example, $\widetilde{e_0 e_{012\ldots n}} = 0$. For a bijective dual that yields the complementary indices on basis elements, a different definition is needed. Following Dorst [17], we use the right complement. This involves choosing an orthogonal basis and then for a basis $k$-vector $x$ to define the dual $x^*$ to be the basis $n+1-k$-vector such that $x \wedge x^* = \mathcal{I}$, for pseudoscalar $\mathcal{I} = e_{012\ldots n}$. For example, this gives dual $e_{01}^* = e_{23}$, so that $e_{01} \wedge e_{23} = e_{0123}$.

This dual is still easy to compute numerically, but it can no longer be constructed solely from operations available to us in the geometric algebra. This makes it more difficult to reason about equivariance.

**Proposition 7.** *In the algebra* $\mathbb{G}_{n,0,1}$, *the join* $a \vee b = (a^* \wedge b^*)^{-*}$ *is equivariant to* $\mathrm{Spin}(n,0,1)$.

*Proof.* Even though the dual is not a $\mathbb{G}_{n,0,1}$ operation, we can express the join in the algebra as follows. We decompose a $k$-vector $x$ as $x = t_x + e_0 p_x$ into a Euclidean $k$-vector $t_x$ and a Euclidean $(k-1)$-vector $p_x$. Then Dorst [17, Eq (35)] computes the following expression

$$(t_x + e_0 p_x) \vee (t_y + e_0 p_y) = ((t_x + e_0 p_x)^* \wedge (t_y + e_0 p_y)^*)^{-*}$$
$$= t_x \vee_{\mathrm{Euc}} p_y + (-1)^n \widehat{p_x} \vee_{\mathrm{Euc}} t_y + e_0 (p_x \vee_{\mathrm{Euc}} p_y), \quad (10)$$

where the Euclidean join of vectors $a, b$ in the projective algebra is defined to equal the join of the corresponding vectors in the Euclidean algebra:

$$a \vee_{\mathrm{Euc}} b := ((a\,\widetilde{e_{12\ldots n}}) \wedge (b\,\widetilde{e_{12\ldots n}}))e_{12\ldots n}$$

The operation $a \vee_{Euc} b$ is $\mathrm{Spin}(n,0,0)$ equivariant, as discussed in Lemma 5. For any rotation $r \in \mathrm{Spin}(n,0,1)$ (which is Euclidean), we thus have $r[a \vee_{\mathrm{Euc}} b] = r[a] \vee_{\mathrm{Euc}} r[b]$. This makes the PGA dual in Eq. (10) equivariant to the rotational subgroup $\mathrm{Spin}(n,0,0) \subset \mathrm{Spin}(n,0,1)$.

We also need to show equivariance to translations. Let $v$ be a Euclidean vector and $\tau = 1 - e_0 v/2$ a translation. Translations act by shifting with $e_0$ times a contraction: $\tau[x] = x - e_0(v \rfloor x)$. This acts on the decomposed $x$ in the following way: $\tau[t_x + e_0 p_x] = \tau[t_x] + e_0 p_x = t_x + e_0(p_x - v \rfloor t_x)$.

We thus get:

$$\tau[x] \vee \tau[y] = (\tau[t_x] + e_0 p_x) \vee (\tau[t_y] + e_0 p_y)$$
$$= (t_x + e_0(p_x - v \rfloor t)) \vee (t_y + e_0(p_y - v \rfloor t_y))$$
$$= x \vee y - t_x \vee_{\mathrm{Euc}} (v \rfloor t_y) - (-1)^n \widehat{v \rfloor t_x} \vee_{\mathrm{Euc}} t_y$$
$$\qquad - e_0 \left( p_x \vee_{\mathrm{Euc}} (v \rfloor t_y) + (v \rfloor t_x) \vee_{\mathrm{Euc}} p_y \right) \qquad \text{Used (10) \& linearity}$$
$$= x \vee y - e_0 \left( p_x \vee_{\mathrm{Euc}} (v \rfloor t_y) + (v \rfloor t_x) \vee_{\mathrm{Euc}} p_y \right) \qquad \text{Used (9)}$$
$$= x \vee y - e_0 \left( -(-1)^n \widehat{v \rfloor p_x} \vee_{\mathrm{Euc}} t_y + (v \rfloor t_x) \vee_{\mathrm{Euc}} p_y \right) \qquad \text{Used (9)}$$
$$= x \vee y - e_0 \left( (-1)^n (v \rfloor \widehat{p_x}) \vee_{\mathrm{Euc}} t_y + (v \rfloor t_x) \vee_{\mathrm{Euc}} p_y \right)$$
$$= x \vee y - e_0 \left( v \rfloor \left\{ (-1)^n \widehat{p_x} \vee_{\mathrm{Euc}} t_y + t_x \vee_{\mathrm{Euc}} p_y \right\} \right) \qquad \text{Used (8)}$$
$$= \tau[x \vee y]$$

The join is thus equivariant [12] to translations and rotations and is therefore $\mathrm{Spin}(n,0,1)$ equivariant. $\square$

Similar to the Euclidean case, we obtain full $\mathrm{Pin}(n,0,1)$ equivariance via multiplication with a pseudoscalar. We thus also use the EquiJoin from Eq. (7) in the projective case.

### A.3 Expressivity

As also noted in Ref. [17], in the projective algebra, the geometric product itself is unable to compute many quantities. It is thus insufficient to build expressive networks. This follows from the fact that the geometric product preserves norms.

---

[12]The authors agree with the reader that there must be an easier way to prove this.

**Lemma 7.** *For the algebra* $\mathbb{G}_{n,0,r}$, *for multivectors* $x, y$, *we have* $\|xy\| = \|x\| \, \|y\|$.

*Proof.* $\|xy\|^2 = xy\widetilde{xy} = xy\tilde{y}\tilde{x} = x\|y\|^2\tilde{x} = x\tilde{x}\|y\|^2 = \|x\|^2\|y\|^2$ $\qquad\qquad\square$

Hence, any null vector in the algebra can never be mapped to a non-null vector, including scalars. The projective algebra can have substantial information encoded as null vector, such as the position of points. This information can never influence scalars or null vectors. For example, there is no way to compute the distance (a scalar) between points just using the projective algebra. In the GATr architecture, the input to the MLPs that operate on the scalars, or the attention weights, thus could not be affected by the null information, had we only used the geometric product on multivectors.

To address this limitation, we use besides the geometric product also the join. The join is able to compute such quantities. For example, given the Euclidean vector $e_{12\ldots n}$, we can map a null vector $x = e_{012\ldots k}$ to a non-null vector $x \vee e_{12\ldots n} \propto e_{12\ldots k}$.

# B Architecture

**Equivariant join**  One of the primitives in GATr is the equivariant join $\mathrm{EquiJoin}(x, y; z)$, which we define in Eq. (7). For $x$ and $y$, we use hidden states of the neural network after the previous layer. The nature of $z$ is different: it is a reference multivector and only necessary to ensure that the function correctly changes sign under mirrorings of the inputs. We find it beneficial to choose this reference multivector $z$ based on the input data rather than the hidden representations, and choose it as the mean of all inputs to the network.

**Auxiliary scalars**  In addition to multivector representations, GATr supports auxiliary scalar representations, for instance to describe non-geometric side information such as positional encodings or diffusion time embeddings. In most layers, these scalar variables are processed like in a standard Transformer, with two exceptions. In linear layers, we allow for the scalar components of multivectors and the auxiliary scalars to freely mix. In the attention operation, we compute attention weights as

$$\mathrm{Softmax}_i\left( \frac{\sum_c \langle q_{i'c}^{MV}, k_{ic}^{MV}\rangle + \sum_c q_{i'c}^s k_{ic}^s}{\sqrt{8n_{MV} + n_s}} \right), \tag{11}$$

where $q^{MV}$ and $k^{MV}$ are query and key multivector representations, $q^s$ and $k^s$ are query and key scalar representations, $n_{MV}$ is the number of multivector channels, and $n_s$ is the number of scalar channels.

**Distance-aware dot-product attention**  As we argue in Sec. 3.3, it can be beneficial to extend queries and keys with nonlinear features. We use the following choice:

$$\phi(q) = \omega(q_{\backslash 0})\begin{pmatrix} q_{\backslash 0}^2 \\ \sum_i q_{\backslash i}^2 \\ q_{\backslash 0} \, q_{\backslash 1} \\ q_{\backslash 0} \, q_{\backslash 2} \\ q_{\backslash 0} \, q_{\backslash 3} \end{pmatrix} \quad \text{and} \quad \psi(k) = \omega(k_{\backslash 0})\begin{pmatrix} -\sum_i k_{\backslash i}^2 \\ -k_{\backslash 0}^2 \\ 2k_{\backslash 0} \, k_{\backslash 1} \\ 2k_{\backslash 0} \, k_{\backslash 2} \\ 2k_{\backslash 0} \, k_{\backslash 3} \end{pmatrix} \quad \text{with} \quad \omega(x) = \frac{x}{x^2 + \epsilon} \, .$$

Here the index $\backslash i$ denotes the trivector component with all indices *but* $i$. Then

$$\phi(q) \cdot \psi(k) = -\omega(q_{\backslash 0})\omega(k_{\backslash 0})\|k_{\backslash 0}\, \vec{q} - q_{\backslash 0}\, \vec{k}\|_{\mathbb{R}^3}^2 \, ,$$

where we use the shorthand $\vec{x} = (x_{\backslash 1}, x_{\backslash 2}, x_{\backslash 3})^T$. When the trivector components of queries and keys represent 3D points, with $q_{\backslash 0} = k_{\backslash 0} = 1$, it becomes proportional to the pairwise negative squared Euclidean distance between the points.

In this notation, a trivector $q = (q_{\backslash 0}, \vec{q})$ transforms under rotation $R \in \mathrm{O}(3)$ and translation $\vec{t} \in \mathbb{R}^3$

as $(q_{\backslash 0}, \vec{q}) \mapsto (q_{\backslash 0}, R\vec{q} + q_{\backslash 0}\vec{t})$. It is easy to see that the inner product is invariant:

$$
\begin{aligned}
\phi(q) \cdot \psi(k) &\mapsto -\omega(q_{\backslash 0})\omega(k_{\backslash 0})\|k_{\backslash 0}\,(R\vec{q} + q_{\backslash 0}\vec{t}) - q_{\backslash 0}\,(R\vec{k} + k_{\backslash 0}\vec{t})\|_{\mathbb{R}^3}^2 \\
&= -\omega(q_{\backslash 0})\omega(k_{\backslash 0})\|R(k_{\backslash 0}\,\vec{q} - q_{\backslash 0}\,\vec{k}) + q_{\backslash 0}k_{\backslash 0}\vec{t} - q_{\backslash 0}k_{\backslash 0}\vec{t}\|_{\mathbb{R}^3}^2 \\
&= -\omega(q_{\backslash 0})\omega(k_{\backslash 0})\|k_{\backslash 0}\,\vec{q} - q_{\backslash 0}\,\vec{k}\|_{\mathbb{R}^3}^2 \\
&= \phi(q) \cdot \psi(k)\,.
\end{aligned}
$$

The normalization function $\omega$ could have been chosen as $\omega(x) = 1$, but this would make the attention logit a fourth-order polynomial of the keys and queries, which tends to lead to instabilities when taking the exponential in the softmax. The choice above makes the attention logit a quadratic function for larger values of the keys and queries, just like regular dot-product attention.

For the interested reader, we'd like to note that the above inner product is closely related to the inner product in another geometric algebra, the conformal geometric algebra. We plan on exploring this connection in future work.

With this final piece, GATr's attention mechanism computes attention weights from three sources: the $\mathbb{G}_{3,0,1}$ inner product of the multivector queries and keys $\langle q, k \rangle$, the distance-aware inner product of the nonlinear features $\phi(q) \cdot \psi(k)$, and the Euclidean inner product of the auxiliary scalars $q_s \cdot k_s$.

We find it beneficial to add learnable weights as prefactors to each of these three terms. The attention weights are then given by

$$
\text{Softmax}_i\left( \frac{\alpha \sum_c \langle q_{i'c}^{MV}, k_{ic}^{MV} \rangle + \beta \sum_c \phi(q_{i'c}^{MV}) \cdot \psi(k_{ic}^{MV}) + \gamma \sum_c q_{i'c}^s k_{ic}^s}{\sqrt{13 n_{MV} + n_s}} \right)
$$

with learnable, head-specific $\alpha, \beta, \gamma > 0$.

This may look complicated, but all terms can be summarized in a single Euclidean dot product between query features and key features. We can therefore use efficient implementations of dot-product attention to compute GATr's attention.

**Multi-query attention** To reduce memory use, we also consider a version of GATr that uses multi-query attention [43] instead of multi-head attention, sharing the keys and values among attention heads.

## C   Experiments

### C.1   $n$-body dynamics

**Dataset** We generate a synthetic $n$-body dataset for $n$ objects by following these steps for each sample:

1. The masses of $n$ objects are sampled from log-uniform distributions. For one object (the star), we use $m_0 \in [1, 10]$; for the remaining objects (the planets), we use $m_i \in [0.01, 0.1]$. (Following common practice in theoretical physics, we use dimensionless quantities such that the gravitational constant is 1.)
2. The initial positions of all bodies are sampled. We first use a heliocentric reference frame. Here the initial positions of all bodies are sampled. The star is set to the origin, while the planets are sampled uniformly on a plane within a distance $r_i \in [0.1, 1.0]$ from the star.
3. The initial velocities are sampled. In the heliocentric reference frame, the star is at rest. The planet velocities are determined by computing the velocity of a stable circular orbit corresponding to the initial positions and masses, and then adding isotropic Gaussian noise (with standard deviation 0.01) to it.
4. We transform the positions and velocities from the heliocentric reference frame to a global reference frame by applying a random translation and rotation to it. The translation is sampled from a multivariate Gaussian with standard deviation 20 and zero mean (except for the domain generalization evaluation set, where we use a mean of $(200, 0, 0)^T$). The rotation is sampled from the Haar measure on $SO(3)$. In addition, we apply a random permutation of the bodies.

5. We compute the final state of the system by evolving it under Newton's equations of motion, using Euler's method and 100 time steps with a time interval of $10^{-4}$ each.
6. Finally, samples in which any bodies have traveled more than a distance of 2 (the diamater of the solar system) are rejected. (Otherwise, rare gravitational slingshot effects dominate the regression loss and all methods become unreliable.)

We generate training datasets with $n = 4$ and between 100 and $10^5$ samples; a validation dataset with $n = 4$ and 5000 samples; a regular evaluation set with $n = 4$ and 5000 samples; a number-generalization evaluation set with $n = 6$ and 5000 samples; and a E(3) generalization set with $n = 4$, an additional translation (see step 4 above), and 5000 samples.

All models are tasked with predicting the final object positions given the initial positions, initial velocities, and masses.

**Models**   For the GATr model, we embed object masses as scalars, positions as trivectors, and velocities (like translation vectors) as bivectors. We use 10 attention blocks, 16 multivector and 128 scalar channels, and 8 attention heads, resulting in 1.9 million parameters. We use multi-head attention and do not use the distance-aware attention mechanism (as it led to similar results in a short test).

For the Transformer baseline, we follow a pre-layer normalization [2, 54] architecture with GELU activations [23] in the MLP block. We use 10 attention blocks, 384 channels, and 8 attention heads, resulting in 11.8 million parameters.

We also compare to a simple MLP with GELU activations. We test models with 2, 5, and 10 layers, each with 384 channels. We report results for the best-performing 2-layer MLP, which has 0.2 million parameters.

Next, we compare GATr to the geometric-algebra-based, but not equivariant, GCA-GNN [41]. We follow the hyperparameters reported by Ruhe et al. [41] for their Tetris experiment, arriving at an architecture with 1.5 million parameters. We also experiment with the GCA-MLP architecture [41], but find worse results.

Finally, we compare to two equivariant baselines: SEGNN [6] and the SE(3)-Transformer [20]. In both cases we use the code released by the authors and the hyperparameters they recommend for (slightly different) $n$-body experiments. This leads to models with 0.1 (SEGNN) and 0.3 (SE(3)-Transformer) million parameters. For SEGNN, we vary the number of nearest neighbours between 3 and the number of objects in the scene (corresponding a fully connected graph) and show the best result.

**Training**   All models are trained by minimizing a $L_2$ loss on the final position of all objects. We train for $50\,000$ steps with the Adam optimizer, using a batch size of $64$ and exponentially decaying the learning rate from $3 \cdot 10^{-4}$ to $3 \cdot 10^{-6}$.

### C.2    Arterial wall-shear-stress estimation

**Dataset**   We use the single-artery wall-shear-stress dataset published by Suk et al. [48]. It consists of 2000 meshes of human arteries, of which 1600 are used for training the remaining for validation and evaluation. Each mesh has around 7000 nodes.

We experiment both with randomly rotated meshes as well as a canonicalized version, in which all meshes point in the same direction.

**Models**   For the GATr model, we embed object the positions of the mesh nodes as trivectors, the mesh surface normals as vectors, and the geodesic distance to the inlet as scalars. We use multi-query attention, the distance-aware attention mechanism, and learnable prefactors for the attention weights. Our model has 10 attention blocks, 8 multivector channels and 32 scalar channels, and 4 attention heads, resulting in 0.2 million parameters.

For the Transformer baseline, we follow a pre-layer normalization [2, 54] architecture with GELU activations [23] in the MLP block. We use 10 attention blocks, 160 channels, and 4 attention heads, resulting in 1.7 million parameters.

In addition, we compare to the results reported by Suk et al. [48].

| Method | Mean approximation error [%] | |
| --- | --- | --- |
| | Random orientations | Canonical orientations |
| GATr | **5.6** | **5.5** |
| Transformer | 10.5 | 6.0 |
| PointNet++ [36] | 12.3 | 8.6 |
| GEM-CNN [15] | 7.7 | 7.8 |

Table 2: Arterial wall-shear-stress estimation [48]. We show the mean approximation error (lower is better), reporting results both on randomly oriented training and test samples (left) and on a version of the dataset in which all artery meshes are canonically oriented (right). We compare GATr to our own Transformer baseline as well as to the results reported by Suk et al. [48].

| Method | Mean approx. error [%] |
| --- | --- |
| GATr (default) | **6.1** |
| Without multivector representations (= Transformer) | 10.5 |
| Without aux. scalar representations | 10.4 |
| Without equivariance constraints on linear maps | 9.7 |
| Less wide (4 MV channels + 8 scalar channels) | 12.0 |
| Less deep (5 blocks) | 7.8 |

Table 3: Ablation experiments on the arterial wall-shear-stress dataset. We report the mean approximation error (lower is better) on the validation set after training for 100 000 steps.

**Training**    We train the models on a $L_2$ loss on wall shear stress. We train for 200 000 steps with the Adam optimizer, using a batch size of 8 and exponentially decaying the learning rate from $3 \cdot 10^{-4}$ to $3 \cdot 10^{-6}$.

To fit the models into memory, we use mixed 16-bit / 32-bit precision, gradient checkpointing [9], and multi-query attention. Most importantly, we compute dot-product attention with the implementation by [32].

**Results**    The results of our experiments are shown in Fig. 3 and in Tbl. 2.

**Ablation studies**    We also perform ablation experiments to study the impact of multiple design decisions on the performance. All major design choices appear to play an important role: without multivector representations, without auxiliary scalar representations, or without equivariance constraints, GATr performs much worse. Further reducing the model's width also severely harms the performance.

### C.3    Robotic planning through invariant diffusion

**Environment**    We use the block stacking environment from Janner et al. [27]. It consists of a Kuka robotic arm interacting with four blocks on a table, simulated with PyBullet [13]. The state consists of seven robotic joint angles as well as the positions and orientations of the four blocks. We consider the task of stacking four blocks on top of each other in any order. The reward is the stacking success probability and is normalized such that 0 means that no blocks are ever successfully stacked, while 100 denotes perfect block stacking.

**Dataset and data parameterization**    We train models on the offline trajectory dataset published by Janner et al. [27]. It consists of 11 000 expert demonstrations.

To facilitate a geometric interpretation, we re-parameterize the environment state into the positions and orientations of the robotic endeffector as well as the four blocks. The orientations of all objects are given by two direction vectors. In addition, there are attachment variables that characterize whether the endeffector is in contact with either of the four blocks. In this parameterization, the environment state is 49-dimensional.

We train models in this geometric parameterization of the problem. To map back to the original parameterization in terms of joint angles, we use a simple inverse kinematics model that solves for the joint angles consistent with a given endeffector pose.

| Method | Reward |
|---|---|
| GATr-Diffuser (ours) | **74.8** $\pm 1.7$ |
| Transformer-Diffuser | $69.8 \pm 1.9$ |
| Diffuser [27] (reproduced) | $57.7 \pm 1.8$ |
| Diffuser [27] | $58.7 \pm 2.5$ |
| EDGI [7] | $62.0 \pm 2.1$ |
| CQL [30] | 24.4 |
| BCQ [21] | 0.0 |

Table 4: Diffusion-based robotic planning. We show the normalized cumulative rewards achieved on a robotic block stacking task [27], where 100 is optimal and means that each block stacking task is completed successfully, while 0 corresponds to a failure to stack any blocks. We show the mean and standard error over 200 evaluation episodes. The top three results were computed in the GATr code base, the bottom four taken from the literature [7, 27].

**Models**  For the GATr model we embed object positions as trivectors, object orientations as oriented planes, gripper attachment variables as scalars, and the diffusion time as scalars. We use axial attention, alternating between attending over time steps and over objects, with positional embeddings along the time axis. Neither multi-query attention nor the distance-aware attention mechanism are used. We train three models, with 10, 20, and 30 Transformer blocks. In each case we use 16 multivector plus 32 scalar channels and 8 attention heads. We report results for the best-performing model with 20 Transformer blocks and 4.0 million parameters.

For the Transformer baseline, we also use axial attention, pre-layer normalization, GELU activations, and rotary positional embeddings. We experiment with 6 models, with 10, 20, or 30 Transformer blocks and 144 or 384 channels. All use 8 attention heads. We report results for the best model, which has 20 Transformer blocks, 144 channels, and 3.5 million parameters.

For the Diffuser baseline, we follow the architecture and hyperparameters described by Janner et al. [27]. The model has 65.1 million parameters.

All models are embedded in a diffusion pipeline as described by Ho et al. [25], using the diffusion time embedding and hyperparameter choices of Ref. [27]. In particular, we use univariate Gaussian base densities and 1000 diffusion steps.

**Training**  We train all models by minimizing the simplified diffusion loss proposed by Ho et al. [25]. For our GATr models and the Diffuser baselines we use an $L_2$ loss and train for $200\,000$ steps with the Adam optimizer, exponentially decaying the learning rate from $3 \cdot 10^{-4}$ to $3 \cdot 10^{-6}$. This setup did not work well for the Diffuser model, where (following Janner et al. [27]) we use a $L_1$ loss and a low constant learning rate instead.

**Evaluation**  All models are evaluated by rolling out at least 200 episodes in a block stacking environment and reporting the mean task and the standard error. We use the planning algorithm and parameter choices of Janner et al. [27] (we do not optimize these, as our focus in this work is on architectural improvements). It consists of sampling trajectories of length 128 from the model, conditional on the current state; then executing these in the environment using PyBullet's PID controller. Each rollout consists of three such phases.

**Results**  The results of our experiments are shown in Fig. 4 and in Tbl. 4.

## C.4  Scaling

**Setup**  To study computational requirements and scaling behaviour, we measure the wall time and peak GPU memory usage during a combined forward and backward pass for various models. We use random Gaussian input data with a batch size of 4, a variable number of items (tokens), and four input channels. After a number of warmup steps, measure the resource use over ten successive steps and report the average time and peak memory usage. For all measurements, we use mixed 16-bit / 32-bit precision, gradient checkpointing, and an efficient attention implementation by Lefaudeux et al. [32].

**Models**   The GATr model has 10 blocks, 8 multivector and 16 scalar channels, and 4 attention heads. We use multi-query attention, the distance-aware attention mechanism, and learnable prefactors for the attention weights.

For the baselines, we choose hyperparameters to match GATr in terms of depth (number of blocks), width (total size of hidden channels), and token connectivity (allowing all tokens to interact with each other in each layer). For the Transformer, this leads to the same choices as for GATr, except that 144 hidden channels are used.

For SEGNN and the $SE(3)$-Transformer, we use fully connected graphs, $O(3)$ representations $\ell \leq 1$, a total of 144 hidden features distributed between scalar and vector representations, and 10 layers. We use the official implementation by Brandstetter et al. [6] for SEGNN and the optimized implementation by Milesi [35] for the $SE(3)$-Transformer.

