# OpenReview forum: "Geometric Algebra Transformer"
_NeurIPS.cc/2023/Conference — NeurIPS 2023 poster_

### Official Review · Reviewer_qqQC · 2023-07-02

**Soundness:** 3 good
**Presentation:** 2 fair
**Contribution:** 3 good
**Rating:** 6
**Confidence:** 2

**Summary:**

This paper is about geometric algebra, which is interesting but not widely known. In this paper, the authors propose a transformer-based model for geometric data, which can represent general geometric data and transformation. The authors also apply this method in two cases, which shows the effectiveness of the proposed method.

**Strengths:**

(1)	The research topic is interesting and useful, because the geometric data is widely existed in the application.

(2)	The experimental results seem good.


**Weaknesses:**

(1)	Although the authors give some background in this paper, it is still not friendly for readers with little knowledge in geometric algebra.

(2)	In this paper, there is not any ablation study.


**Questions:**

(1)	In the section 4.1, it should be Fig.3 instead of Fig.1. The authors can check this.

(2)	In Fig.3, it seems the number of training samples does not affect the performance of GATr, why?


**Limitations:**

From the Figure 1, it seems we need to obtain the geometric types from a processing block. I wonder how the accuracy of this block.

---

> ### Author Rebuttal · Authors · 2023-08-09
>
> Thank you for the constructive review and helpful questions.
>
>
> > There is no ablation study.
>
> Thank you for the suggestion. We ran an ablation study on the main architectural decisions in GATr.
>
> We performed this experiment on a new real-world problem: estimating the wall shear stress from artery meshes. We discuss this dataset in the overall response and show results on it on the rebuttal result page.
>
> Here we provide some additional ablation results. We report the normalized mean approximation error on the validation set after training for 100k steps:
>
> | Method | approximation error [%] |
> | --- | --- |
> | GATr (standard) | **6.1** |
> | Without multivector reps | 10.5 |
> | Without scalar reps | 10.4 |
> | Without equivariance constraints | 9.7 |
> | Less wide | 12.0 |
> | Less deep | 7.8 |
>
> We conclude that all the main design choices in GATr are important for the strong performance on this dataset, and we cannot reduce the capacity much from the run. We have not yet systematically tested increasing the model capacity. We will add an extended version of this ablation study to the final version of our paper.
>
>
> > In the section 4.1, it should be Fig. 3 instead of Fig. 1.
>
> Thanks for catching this, we will fix it.
>
>
> > In Fig. 3, it seems the number of training samples does not affect the performance of GATr.
>
> It is hard to see in this log-scale plot, but GATr's error does actually decrease with training set size, from 6e-4 to 4e-4 mean absolute error. But we were also surprised to see how strong GATr performs even with very little training data. This is testament to the strong geometric inductive bias from equivariance and geometric algebra representations.
>
>
> > From the Figure 1, it seems we need to obtain the geometric types from a processing block. I wonder how the accuracy of this block.
>
> This is a great question, and the answer will depend on the application. In our experiments, we assume knowledge of positions and/or momenta of the objects, so this is no source of errors.
>
> In contrast, real-world robotics applications will require estimating the positions and poses of objects from sensor inputs, which will lead to some approximation error. We believe that such errors can be quite low when e.g. depth cameras are used. We are planning to study integrating this preprocessing step into the pipeline in future work.
>
>
> Thank you again for the review. We hope that we could address your questions and look forward to discussing further. Please also have a look at our overall response, where we present some new results. In addition to the artery experiment, we also compare GATr to more equivariant baselines. Finally, we study its computational requirements and scaling behaviour, showing that GATr can be scaled to much larger systems than equivariant baselines.

---

### Official Review · Reviewer_yyFc · 2023-07-04

**Soundness:** 4 excellent
**Presentation:** 3 good
**Contribution:** 4 excellent
**Rating:** 8
**Confidence:** 3

**Summary:**

This paper introduces the Geometric Algebra Transformer, a novel architecture that leverages the projective geometric algebra to embed inputs, outputs, and hidden features. The transformer incorporates specific linear layers, nonlinear layers, normalization layers, and attention mechanisms designed for the projective geometric algebra. As a general equivariant architecture, the proposed transformer demonstrates significant improvements in experiments, highlighting its effectiveness compared to other models that do not consider geometric information in their features.

**Strengths:**

1. This paper stands out as the first to integrate projective geometric algebra into an equivariant network based on $E(3)$. Unlike previous $E(n)$ networks, the utilization of projective algebra enables the representation of a wider range of geometric elements such as data points, rays, planes, and transformations. The design of the required transformer layers is both straightforward to implement and versatile, facilitating scalability and offering numerous applications in machine learning, computer vision, and robotics.

2. The proposed method is the first equivariant networks with respect to $E(3)$, in contrast with the previous models dealing with $O(3)$ equivariance.

3. The experimental results are impressive, showcasing significant performance enhancements and efficient training data utilization. These outcomes underscore the efficacy and potential of the proposed model.

4. Despite the complexity associated with projective geometric algebra, the paper effectively conveys its ideas through clear explanations and illustrative figures. The authors ensure that readers without prior knowledge of projective algebra can grasp the general concepts and insights presented in the work.

**Weaknesses:**

1. The proposed model's requirement of 16-dimensional vectors for feature representation may present scalability challenges compared to traditional transformers. Similar to previous equivariant networks utilizing different types of features, this can lead to memory overhead and increased computational complexity. Further experimental analysis in this regard would be valuable to better understand the implications and potential trade-offs.

2. Correct if I'm wrong. When the model adopts the rotary positional embeddings, I think it cannot contain the permutation and rotation equivariance since the transformation will change the corresponding positional embeddings.

3. The experiments might lack one baseline, that is the $SE(3)$ equivariant transformer. I think it would be beneficial to provide the insights of the priority of projective geometric algebra representation.

**Questions:**

1. Is there any theoretical relation or comparison between the GATrs and the previous $E(3)$ equivariant networks with steerable features, since they all consider the geometric information in the network.

2. Would there any future directions to measure the expressiveness of GATrs?

3. For the nonlinear layers, would the nonliearity acting on scalar components or the norm impact the expressivity of the network, and does there exist other nonlinearities like the previous approaches in $E(3)$ equivariant network?


**Limitations:**

The authors addressed the limitations in the paper.

---

> ### Author Rebuttal · Authors · 2023-08-09
>
> Thank you for the in-depth review, kind praise, and on-point questions.
>
>
> > The proposed model's requirement of 16-dimensional vectors for feature representation may present scalability challenges compared to traditional transformers.
>
> Good point. To address it, we added a study of the computational requirements and scalability of GATr. Please have a look at the overall response for the results and a discussion.
>
> In a nutshell, we find that GATr indeed has some overhead in terms of compute time compared to a regular Transformer when working with problems with a small or moderate number of tokens. However, the scaling behaviour to many tokens is the same as that of a Transformer, as both architectures are bottlenecked by the same dot-product attention mechanism. When using an efficient implementation of dot-product attention, we are able to **scale GATr to tens of thousands of tokens**, while equivariant baselines like SEGNN [6] or SE(3)-Transformer [19] run out of memory much sooner.
>
>
> > When the model adopts the rotary positional embeddings, I think it cannot contain the permutation and rotation equivariance.
>
> We need to apologize for this confusing terminology, which we carelessly adapted from non-geometric transformers. "Position" here means position in a sequence, not geometric position in 3D. "Rotary" does not refer a rotation of 3D space, but rather to how the position in a sequence is embedded via sinusoids in the *scalar* channels of keys and queries. Using positional encoding thus does not affect rotational equivariance. We will use better terminology in the final version.
>
> Positional embeddings do indeed break permutation equivariance. We only use them to indicate the position in a sequence when this is desired. In our experiments, this is only the case for the time dimension of the robotic manipulation problem.
>
>
> > The experiments might lack one baseline, that is the $SE(3)$ equivariant transformer.
>
> Thank you for the suggestion. We added the **SE(3)-Transformer** to our n-body experiment. For good measure, we also added another popular equivariant method, the **SEGNN** [6]. Please have a look at our overall response and the rebuttal result page.
>
> Both the SE(3)-Transformer and SEGNN perform on par with GATr when trained on enough samples, but GATr is more sample-efficient, outperforming both baselines when training on little data. This provides further evidence that not only equivariance, but also the geometric algebra representations provide a useful inductive bias for geometric problems.
>
>
> > Is there any theoretical relation or comparison between the GATrs and the previous $E(3)$ equivariant networks with steerable features?
>
> There are two key differences between GATr and prior work, like SEGNN [6]. Firstly, we use a geometric algebra as a representation, rather than E(3) irreducible representations. This is on the one hand a richer representation, as it can represent the E(3) group faithfully having features that transform (non-trivially) under translations. On the other hand, when decomposing the algebra into O(3) irreps, one sees that it does not contain irreducible representation of frequency higher than 1, which are often used in other works.
>
> A second difference is that we compute pair-wise interaction via attention, which in effect just takes linear combinations of representations weighted by an invariant scalar. This is different from SEGNN, which uses a neural network to compute pairwise interaction. It remains to be seen theoretically how limiting these choices are, but as we write in our overall response, we have good reasons to believe this does not harm GATr's expressivity.
>
> > Would there be any future directions to measure the expressiveness of GATrs?
>
> Thank you for the suggestion. We address it in our overall response. In short, while we do not have a formal universality theorem, we have some arguments and empirical evidence that GATr is fully expressive. We aim to develop a formal result in future work.
>
> > Would the nonlinearity acting on scalar components or the norm impact the expressivity of the network? [...] Are there other nonlinearities like the previous approaches in $E(3)$ equivariant network?
>
> Equivariant neural networks with infinite groups (like E(3)) typically use three kinds of non-linearities: (1) polynomials via tensor products [52], (2) multipling a reprensentation by a scalar (derived from its norm or a scalar gate) [53], (3) approximate equivariance by sampling to a permutation representation, then using a point-wise non-linearity [54]. GATr uses the first two: the geometric product is a polynomial akin to tensor products and we gate the multivectors by multiplication with a scalar (on which we can equivariantly apply point-wise non-linearities like GELU). Therefore, we believe our choice of non-linearities is very similar to mainstream equivariant networks. We thus see no reason why this should hinder our expressivity.
>
>
> Thank you again for the review. We hope that we could address your questions and look forward to discussing further. Please also have a look at our overall response. In addition to the scaling analysis and the comparison to more baselines, we also test GATr on a arterial shear stress estimation problem, a realistic benchmark with large geometric structures (meshes of 7000 nodes) and small training sets (1600 training samples). Here GATr sets a new state of the art.
>
> **References**
>
> See paper, plus:
>
> [52] R. Kondor et al., "Clebsch-Gordan Nets: a Fully Fourier Space Spherical Convolutional Neural Network", NeurIPS 2018
>
> [53] M. Weiler et al., "3D Steerable CNNs: Learning Rotationally Equivariant Features in Volumetric Data",  NeurIPS 2018
>
> [54] T. Cohen el al., "Spherical CNNs", ICLR 2018

---

### Official Review · Reviewer_wqKZ · 2023-07-05

**Soundness:** 3 good
**Presentation:** 3 good
**Contribution:** 3 good
**Rating:** 6
**Confidence:** 2

**Summary:**

The paper proposes a (transformer-based) network architecture that operates on a projective geometric algebra that can encode simple 3D transformations and 3D objects (such as points, planes, etc) in a unified description. The network performs operations on these mathematical objects that are (also in composition) equivariant, leading to a frame-invariant computation that is desired for many geometric applications of deep learning, while the projective approach extends the set of transformations that can be employed to cover all of E(3). The paper shows two prototypical examples with geometric learning in gravitational particle simulation and planning of the motion of a robot arm.
The main improvement over previous work is the fully equivariant formulation.

**Strengths:**

The paper is well written, easy to read and self-contained despite building on a larger body of previous work with mathematical concepts that are not "mainstream" (yet).

The basic idea appears to be sound and useful: The ability to perform computations on a somewhat richer set of geometric objects and transformations is probably a useful building block for geometric learning algorithms. The specific architecture ensures frame-invariant computations, which is required for many applications.

**Weaknesses:**

One weakness of the proposed approach is the limited invariance wrt to permutations, see "limitations" below. Similarly, it is not clear in how far expressiveness is restricted by the restriction to the specific operations employed.

The paper is also somewhat incremental in nature, as it adds a selection of specific operations that preserve equivariance, but some of the techniques required appear to be non-obvious.

The examples are rather basic; they provide a nice proof-of-concept but do not yet show whether the approach is competitive in more realistic problem scenarios or competitive benchmarks.

Due to the limited practical results and some conceptual limitations I would give only a weakly positive recommendation for this (otherwise) well-rounded submission.

**Questions:**

In how far could permutation invariance (or the lack thereof) become an issue, for example in a complex molecular dynamics simulation with heterogeneous atomistic components and interactions or more complex tasks (e.g. predicting forces or electron densities)? In general, could the presented design perform the same tasks as a (steerable) group-convolutional architecture?

Does the design come with potential limitations in expressiveness over non-geometric networks? (Appendix A.3 and Section 3.2 show that there are some improvements over prior approaches, but are there some known negative results for the actual/novel approach?)

**Limitations:**

Networks that obtain equivariance by employing an algebra of covariant and invariant operations are an established idea for simpler sets of geometric objects, such as collections of tensor fields of different type (e.g., scalars and vectors). The fundamental limitation of this approach is the lack of permutation invariance: As every object receives a specific, learned computation, it becomes non-trivial to achieve permutational and geometric invariance at the same time. Solutions often require substantial architectural restrictions (such as restrictions to commutative uniform operations such as pooling) that might reduce expressivity. In the case of the transformer architecture, the transformer is by design permutationally invariant under its input tokens, but problems arise once multiple geometric objects are embedded into the same token (these cannot be easily modeled as indistinguishable). I am lacking experience with transformer architectures to see in how far this might be an issue for typical applications, but it is probably worth discussing.

It might also be worth considering a bit more in depth in how far expressiveness is constrained by the choice of equivariant operations allowed.

---

> ### Author Rebuttal · Authors · 2023-08-09
>
> Thank you for the detailed and constructive review as well as the kind words.
>
>
> > In how far could permutation invariance (or the lack thereof) become an issue?
>
> Our architecture is fully **permutation equivariant**: a permutation of the input tokens leads to the same permutation of the output tokens. In this aspect it is exactly like a standard Transformer. We will emphasize this property more in the final version of the paper.
>
> If permutation equivariance is *not* desired in a problem, this equivariance can be broken on the level of the inputs, for instance through positional embeddings. We make use of this option in the robotic manipulation experiments, which have two token dimensions: one between the different objects (along which we maintain permutation equivariance), and one over time steps (here we use positional embeddings to break the permutation symmetry).
>
>
> > Problems arise once multiple geometric objects are embedded into the same token.
>
> Indeed. This is why, in our experiments, we always embed different objects into different tokens. Different properties of the same object (like position, orientation, and velocity of the same object) are embedded into different channels of the same token. We will clarify the embeddings in the appendix.
>
>
> > Does the design come with potential limitations in expressiveness over non-geometric networks?
>
> This is a great question. We address it in our overall response. In short, while we do not have a formal universality theorem, we have some arguments and empirical evidence that GATr is fully expressive.
>
>
> > In general, could the presented design perform the same tasks as a (steerable) group-convolutional architecture?
>
> We believe it can. Based on the theoretical and empirical arguments we give in our overall response, we believe that GATr is expressive enough to represent any equivariant map, just like group-convolutional architectures.
>
> In some sense, our empirical test of expressiveness explicitly tests this: it confirms that GATr can represent any randomly initialized SEGNN network [6], which can be written as group convolutions. Please have a look at our overall response for details.
>
>
> > The paper is also somewhat incremental in nature.
>
> We politely disagree. Not only is our paper the first to propose equivariant networks for projective geometric algebra representations, it also has two properties that are missing from almost all geometric deep learning works.
>
> First, it has **faithful translational E(3) representations**: we can represent points, which transform under translation, not only relative vectors, which are invariant to translation.
>
> Second, it computes pairwise interactions through dot-product attention, which turns out to be much more scalable than the message-passing operations in most equivariant architectures.
>
> We study the scaling behaviour in detail in a new experiment and find that **we can scale GATr to 100 times as many tokens** as SEGNN or the SE(3)-Transformer; see the overall response for a discussion and the rebuttal result page for results.
>
>
> > The examples are rather basic.
>
> We agree, and added a **new experiment on a more realistic problem**: wall shear stress estimation on artery meshes. This is a challenging problem because the meshes are large (**7000 nodes**) and the datasets small (1600 training samples). GATr has no problem scaling to this problem and outperforms all baselines on this task. Please have a look at the overall response for a discussion and the rebuttal result page for the results.
>
>
> Thank you again for the review. We hope that we could address your concerns and look forward to discussing further. Please also have a quick look at our overall response. In addition to the scaling and artery experiments, we compare GATr to more equivariant baselines.

---

> > ### Comment · Reviewer_wqKZ · 2023-08-14
> >
> > Thanks for the detailed feedback. I found the additional example particularly helpful, as it seems to address a more "real-world" application scenario were the new method shines.
> >
> > In terms of the conceptual discussion, I am not very convinced by theoretical expressiveness. Being able to represent any function does not imply that the representation is efficient (thinking of the universality of very shallow networks) nor that solutions are discovered during learning. I found the empirical argument much more convincing (it actually works very well in practice, and compares favorably with GCN-alikes).
> >
> > In terms of having to use separate transformer tokens for representing interchangable objects it is still not clear to me how that affects practical modeling of networks, but that might be due to limited experience with building transformers (which in general exploit symmetry differently from CNNs, it seems).
> >
> > Also the "proper" incorporation of translational invariance is a strong point that escaped me when reading the paper.

---

> > > ### Author Response · Authors · 2023-08-14
> > >
> > > Thank you for the response. We are happy to hear that our additional experiments were convincing, and that you, like us, find the proper translational representations a strong point of our architecture.
> > >
> > > > In terms of having to use separate transformer tokens for representing interchangable objects it is still not clear to me how that affects practical modeling of networks.
> > >
> > > Is it a useful analogy that transformers are like graph networks with a fully connected graph [55]? The tokens in the data are equivalent to the nodes in the graph. As long as the order in which tokens / nodes are stored does not affect the outputs of the network, we have the desired permutation equivariance.
> > >
> > > This is the case for all our network layers. In particular, one can show that our dot-product attention mechanism is permutation-equivariant, just like that in a standard Transformer.
> > >
> > > > Being able to represent any function does not imply that the representation is efficient
> > >
> > > We agree entirely. The ability to universally express any function is in no way a sufficient condition for a good architecture. However, we do see it as more or less a necessary condition, so we do agree with the other reviewers that it may be interesting to study whether GATr satisfies this property.
> > >
> > > Thank you again for your review and the discussion. Is there anything else that you are missing from our paper or that we can clarify? If you found our method and its demonstration convincing and worthy of publication at NeurIPS, we kindly ask you to consider raising your score.
> > >
> > > [55] C. K. Joshi, [Transformers are Graph Neural Networks](https://www.chaitjo.com/post/transformers-are-gnns/), 2021

---

### Official Review · Reviewer_8oHp · 2023-07-11

**Soundness:** 3 good
**Presentation:** 3 good
**Contribution:** 2 fair
**Rating:** 6
**Confidence:** 3

**Summary:**

This paper presents an adaptation of the transformer architecture to enforce equivariance to E(3), which can be helpful to a variety of practical applications necessitating such geometric inductive bias.

This is achieved partly by leveraging geometric algebra rather than vector algebra, which allows for a simple and elegant representation of all 3D sub-spaces from points to volumes and operations within the same mathematical framework via a single product, the geometric product.

The authors review the different operations making up a standard transformer and demonstrate how to adapt them to geometric algebra and bake such equivariance building block by building block (from linear layers, to activation to attention). To illustrate the benefit of this architecture, they take two applications, a n-body simulation and a robotic manipulation where they demonstrate superior performance of the GATr architecture compared to vanilla transformer or MLP (first application) or against a number of baselines include an equivariant planning method.

The specific technical contributions are:
- block-by-block review and adaptation to geometric algebra and E(3) equivariance of the transformer architecture,
- an overall E(3) equivariant architecture readily applicable,
- two applications with demonstrated better performance thanks to this achieved equivariance.


**Strengths:**

The presentation is solid and easy to follow even for someone not necessarily familiar with geometric algebra. That written, a reader unfamiliar with it, will  likely be required to ramp up on the topic via the references to fully grasp the contributions and be able to leverage them.

Geometric algebra is typically a niche mathematical toolkit, which has recently enjoyed a bit more visibility in the graphics community. Shedding more light on this and the practical applications it can have when combined with successful machine learning tools is valuable for the community.

The authors gradually demonstrate and achieve E(3) equivariance with their GATr architecture very methodologically. This architecture can indeed benefit to a number of practical applications that require strong geometric inductive bias.

The proposed evaluations of the architecture, while limited to only two applications, show convincing results with clearly superior performance with the proposed architecture, including (and perhaps more convincingly) against another equivariant baseline in the second application.



**Weaknesses:**

The proposed transformer architecture focuses on achieving E(3) equivariance, enforcing inductive bias that make this architecture particularly suitable to applications with such constraints. However a number of practical applications would require equivariance to SE(3) only rather than E(3), these include fairly important ones in physics, chemistry and biology due to chirality. In the same register, equivariance in 2D would also be rather useful but is not covered in this submission.

Switching to geometric algebra representation carries a significant cost both in terms of storage and computation: each element of the n-D algebra requires 2^n cofficients and the geometric product translates to a 2^n x 2^n matrix. Libraries implementing geometric algebra have to rely on specialization, static or dynamic code generation or caching to achieve computational efficiency closer to standard vector algebra. The authors are touching on this important limitations in the corresponding section of the paper, it would have been relevant to include an assessment of the computational cost in practice in the proposed evaluations (or strategies used to alleviate it).

The submission appears to lack of number of relevant references that achieve some form of equivariance, especially related to the applications evoked in the first weakness above:
- Highly accurate protein structure prediction with AlphaFold - https://www.nature.com/articles/s41586-021-03819-2
    See Methods / IPA section + Supplementary Materials / Invariant point attention (IPA) for SE(3) invariance (a special case of equivariance)
- Cormorant: Covariant Molecular Neural Networks - https://arxiv.org/abs/1906.04015
- Tensor field networks: Rotation- and translation-equivariant neural networks for 3D point clouds - https://arxiv.org/abs/1802.08219
- Equiformer: Equivariant Graph Attention Transformer for 3D Atomistic Graphs - https://arxiv.org/abs/2206.11990
- Equivariant Transformer Networks - https://arxiv.org/abs/1901.11399

It was surprising to not find a mention of equivariance to translation and CNNs. While the second application includes an equivariant baseline, the first one does not. A simple baseline (possibly from the references) with some form of translation (and / or rotation) invariance would strengthen the results of the evaluation, given that the current comparison does not have any geometric equivariance.

**Questions:**

See weaknesses above, could the authors:
- elaborate on the specific choice of equivariance and target applications,
- discuss and illustrate the practical consequences of the efficiency cost on the proposed applications and whether some specialization was used in the applications.

---

> ### Author Rebuttal · Authors · 2023-08-09
>
> Thank you for the thorough review, helpful suggestions, and kind words.
>
>
> > A number of practical applications would require equivariance to SE(3) only. [...] Equivariance in 2D would also be rather useful.
>
> Good point, we agree. There are two strategies to address this: either designing separate architectures for each relevant symmetry subgroup of E(3), or using a universal E(3)-equivariant architecture and breaking the symmetry down to a smaller subgroup through **symmetry-breaking inputs**. In this paper we choose the latter strategy, as we find the universality of the architecture appealing.
>
> Concretely, consider the robotic manipulation experiment. Here the direction of gravity clearly breaks the E(3) symmetry to E(2). We use an E(3)-equivariant architecture, but address the smaller symmetry group by providing a vector pointing in the direction of gravity as part of the network inputs. The results show that that strategy works, but it would indeed be interesting to compare to an E(2)-equivariant architecture.
>
> As another example, relevant e.g. in molecular settings, chirality symmetry can be broken by adding a pseudoscalar input feature, whose sign encodes a choice of orientation.
>
>
> > Each element of the n-D algebra requires 2^n cofficients and the geometric product translates to a 2^n x 2^n matrix
>
> Fair point, and it's even worse: the geometric product is a $2^n \times 2^n \times 2^n$ tensor, though it is quite sparse and can be stored efficiently. Luckily GPUs are very efficient in matrix multiplication, making the overhead manageable.
>
>
> > Include an assessment of the computational cost in practice in the proposed evaluations (or strategies used to alleviate it).
>
> Thanks for the suggestion. We added an experiment measuring the compute time and GPU memory GATr requires as a function of the problem size. We show and discuss the results in the overall response to all reviewers. In short, GATr indeed has some overhead in terms of compute time over a baseline transformer, but scales well to larger problems thanks to the dot-product attention. In particular, GATr **scales much better than the equivariant baselines** SEGNN and SE(3)-Transformer.
>
> Strategies we used to alleviate compute and memory requirements include using efficient implementations of dot-product attention, mixed precision training, gradient checkpointing, and optimized contraction paths in various tensor multiplications (think `einsum`). There is still room for further optimization though, for instance by compilation of the computational graph. We will document all of these tricks and discuss their benefits in the final version of the paper.
>
>
> > The submission appears to lack of number of relevant references that achieve some form of equivariance.
>
> Thank you for the pointers. We will add and discuss these references.
>
>
> We hope that we could address your concerns and look forward to discussing further. Please also have a quick look at our overall response, where we present a few new results. In addition to the scaling experiment, we compare GATr to more equivariant baselines. Finally, we test GATr on a realistic arterial shear stress estimation problem, a benchmark with large geometric structures (meshes of 7000 nodes) and small training sets (1600 training samples). Here GATr sets a new state of the art.

---

> > ### Comment · Reviewer_8oHp · 2023-08-20
> >
> > Thanks to the authors for preparing a thorough rebuttal and taking the time to provide detailed answers to all the reviewers. I have read the overall rebuttal as well as the other reviews and followed the associated discussions.
> >
> > I believe most of my concerns or remarks have been sufficiently addressed by the authors. These answers combined with the scaling experiments and the added realistic application, allows me to bump my overall rating to Weak Accept.

---

### Author Rebuttal · Authors · 2023-08-09

Thanks to the reviewers for their insightful feedback. We are delighted that they find the key ideas behind GATr - geometric algebra representations, equivariance, and a transformer architecture - "sound and useful" (reviewer **wqKZ**) and "very methodological" (**8oHP**). They appreciated both the benefits of true translational E(3) representations and the scalability of the transformer implementation (**yyFc**) and commented positively on the "good" (**qqQC**) experimental results. We are particularly happy that the reviewers found our presentation "well written, easy to read and self-contained" (**wqKZ**).

## Comparison to equivariant baselines

At the suggestion of reviewers **8oHP** and **yyFc**, we added a comparison of GATr to other equivariant architectures on the n-body experiment. We chose the **SE(3)-Transformer** [19], suggested by reviewer **yyFc**, and **SEGNN** [6].

In Fig. 1 of the attached PDF, we show that all three equivariant architectures perform similarly given enough data. However, **GATr is clearly more sample-efficient**.

We also added a comparison to the GCA-GNN architecture [46], also based on projective geometric algebra representations, but not equivariant. GATr clearly outperforms GCA-GNN.

## Expressiveness

Reviewers **wqKZ** and **yyFc** asked about the expressiveness of our architecture. We are interested in theoretically analyzing the expressiveness of GATr as future work, but until such analysis has taken place, we would like to make two observations.

First, Villar et al. [47] show that universal expressiveness of E(3)-equivariant networks can be achieved using merely scalars and vectors, without the need fo higher-order irreps. Scalars and vectors are expressible in the geometric algebra, so we don't think our choice of representation limits expressivity. In addition, we don't think that the transformer architecture is limiting, as they are known to be universal approximators in the non-geometric setting [48].

Second, we studied expressivity empirically. We randomly initialized SEGNN networks [6], which are similar to achitectures known to be universal equivariant approximators [49], and trained GATr networks to imitate them. Our models were always able to reach close to 0 loss. A thorough version of this experiment will be included in the final version.

## Computational cost and scalability

Reviewers **8oHP** and **yyFc** suggested to study the computational costs and scaling behaviour of our architecture. We measured the time and memory requirements of GATr as a function of the number of tokens. We compare to multiple baselines, including a heavily optimized Nvidia implementation of the SE(3)-Transformer [50].

We show the results in Fig. 2 of the rebuttal result page. For few tokens, GATr is slower than a Transformer. However, this difference can be lowered with larger batch sizes or pre-compilation of the computation graph.

For larger problems, compute and memory are dominated by the pairwise interactions in the attention mechanism. Here GATr and the baseline Transformer perform on par, as they use the same efficient implementation of dot-product attention. In terms of memory, both scale linearly in the number of tokens, while the equivariant baselines scale quadratically. In terms of time, all methods scale quadratically, but the equivariant baselines have a worse prefactor than GATr and the Transformer. All in all, **we can easily scale GATr to tens of thousands of tokens**, while the equivariant baselines run out of memory two orders of magnitude earlier.

## Arterial wall shear stress experiment

Reviewer **wqKZ** asked about the performance on more realistic scenarios. We address this with a new experiment on a dataset of synthetic meshes of realistic human arteries [51]. The task is to predict the wall shear stress exerted by the blood flow on the arterial wall, an important predictor of aneurysms. While the wall shear stress can be computed with computational fluid dynamics, simulating a single artery can take many hours, and efficient neural surrogates can have substantial impact. As the meshes are large (around **7000 nodes**) and the datasets small (1600 training meshes), this is a challenging problem.

We show results in Fig. 3 of the attached PDF. Among the methods that do not canonicalize the data, GATr improves upon all previous methods and sets a **new state of the art** on this problem. It outperforms both a Transformer baseline as well as the (equivariant and non-equivariant) results reported by Suk et al. [51].

Canonicalizing the data (rotating arteries such that blood always flows in the same direction) helps the Transformer to be competitive with GATr. However, canonicalization is only feasible for relatively straight arteries as in this dataset, not in more complex scenarios with branching and turning arteries. We find it likely that GATr will be more robust in such scenarios.

## Conclusions

The results we add in this rebuttal highlight the two strengths of GATr: First, its geometric inductive bias allows GATr to perform on par or better than strong equivariant baselines. Second, thanks to its dot-product attention, it scales much more favorably to large problems with thousands of tokens.

## References

See paper, plus:

[46] D. Ruhe et al., "Geometric Clifford Algebra Networks", ICML 2023

[47] S. Villar et al., "Scalars are universal: Equivariant machine learning, structured like classical physics", NeurIPS 2021

[48] C. Yun et al.,"Are Transformers universal approximators of sequence-to-sequence functions?", ICLR 2020

[49] N. Dym & H. Maron, "On the Universality of Rotation Equivariant Point Cloud Networks", arXiv:2010.02449

[50] A. Milesi, "Accelerating SE(3)-Transformers Training Using an NVIDIA Open-Source Model Implementation", 2021

[51] J. Suk et al., "Mesh convolutional neural networks for wall shear stress estimation in 3D artery models", MICCAI 2021

---

> ### Author Response · Authors · 2023-08-19
>
> Thanks again to all reviewers for their constructive feedback on our work.
>
> Since the discussion period is coming to an end, we wanted to respectfully follow up to see if we can still clarify any aspects of GATr.
>
> We believe that in our overall and individual responses we have addressed all major questions and criticisms: following reviewer suggestions, we compared GATr to more equivariant baselines, showed that it can be scaled efficiently to tens of thousands of tokens, and demonstrated it on an additional, realistic problem, where it improves upon the previous state of the art.
>
> If you had time to read our rebuttal, we would greatly appreciate a short response.
> In case we addressed your concerns, we kindly ask you to consider updating your score.
> Should you have any more questions, please let us know; we would be very happy to discuss more.

---

### Decision · Program_Chairs · 2023-09-21

**Decision:**

Accept (poster)

**Comment:**

Thanks for clarifying a few of the loose ends in the rebuttal. All the reviewers liked the submission. Please consider including the relevant bits from the rebuttal in the final version of the paper. Congratulations on a nice piece of work.